# Three-dimensional molecular architecture of mouse organogenesis

Fangfang Qu[1,2,3,9], Wenjia Li[1,3,4,9], Jian Xu[1,9], Ruifang Zhang [1], Jincan Ke[5], Xiaodie Ren[1], Xiaogao Meng[5,6], Lexin Qin[5], Jingna Zhang[1], Fangru Lu[1], Xin Zhou[1], Xi Luo[5], Zhen Zhang[5], Minhan Wang[5], Guangming Wu[1,3,7], Duanqing Pei [8], Jiekai Chen [1,5], Guizhong Cui [1,3,7] ✉, Shengbao Suo [3,4] ✉ & Guangdun Peng [1,5] ✉

Mammalian embryos exhibit sophisticated cellular patterning that is intricately orchestrated at both molecular and cellular level. It has recently become apparent that cells within the animal body display significant heterogeneity, both in terms of their cellular properties and spatial distributions. However, current spatial transcriptomic profiling either lacks three-dimensional representation or is limited in its ability to capture the complexity of embryonic tissues and organs. Here, we present a spatial transcriptomic atlas of all major organs at embryonic day 13.5 in the mouse embryo, and provide a three-dimensional rendering of molecular regulation for embryonic patterning with stacked sections. By integrating the spatial atlas with corresponding single-cell transcriptomic data, we offer a detailed molecular annotation of the dynamic nature of organ development, spatial cellular interactions, embryonic axes, and divergence of cell fates that underlie mammalian development, which would pave the way for precise organ engineering and stem cell-based regenerative medicine.

Organogenesis sets up the functional layout of the animal body from three germ layers. This intricate process involves intensive cell–cell interaction, cell fate determination, cell proliferation, as well as the spatial arrangement of cells into distinct tissues and ultimately functional organs. Recent advancements in technology, such as single-cell transcriptomes and in vitro stem cell-based organoids models, have opened up new avenues for understanding organ development[1]. For instance, systematic single-cell analysis has revealed hundreds of cell types and states and developmental trajectories for many organs

during mouse organogenesis[2]. Similarly, human embryo organogenesis at single-cell resolution has also been reported and major cell types and developmental regulation programs have been delineated[3]. However, the anatomically stringent organization of embryos at organogenesis stages poses challenges for dissociation-based single-cell genomics, as it masks the high-order tissue architecture and location-dependent mechanisms.

Spatially resolved transcriptome technology represents a significant breakthrough in gaining unprecedented views of the

[1]Center for Cell Lineage and Atlas, Bioland Laboratory, Guangzhou, China. [2]GMU–GIBH Joint School of Life Sciences, The Guangdong–Hong Kong–Macau Joint Laboratory for Cell Fate Regulation and Diseases, Guangzhou Medical University, 510005 Guangzhou, Guangdong, China. [3]Guangzhou Laboratory, 510005 Guangzhou, Guangdong, China. [4]The First Affiliated Hospital of Guangzhou Medical University, State Key Laboratory of Respiratory Disease, 510005 Guangzhou, Guangdong, China. [5]Center for Cell Lineage and Development, CAS Key Laboratory of Regenerative Biology, Guangdong Provincial Key Laboratory of Stem Cell and Regenerative Medicine, GIBH-HKU Guangdong–Hong Kong Stem Cell and Regenerative Medicine Research Centre, Guangzhou Institutes of Biomedicine and Health, University of the Chinese Academy of Sciences, Chinese Academy of Sciences, 510530 Guangzhou, China. [6]Life Science and Medicine, University of Science and Technology of China, 230026 Hefei, Anhui, China. [7]School of Basic Medical Sciences, Guangzhou Medical University, 510005 Guangzhou, Guangdong, China. [8]Laboratory of Cell Fate Control, School of Life Sciences, Westlake University, Hangzhou, China. [9]These authors contributed equally: Fangfang Qu, Wenjia Li, Jian Xu. ✉e-mail: cui_guizhong@gzlab.ac.cn; suo_shengbao@gzlab.ac.cn; peng_guangdun@gibh.ac.cn

molecular regionalization of complex tissues and processes. Recently, emerging spatial transcriptome approaches such as Geo-seq[4], seqFISH[5], DBiT-seq[6], Stereo-seq[7], and sci-Space[8], have disentangled the molecular architecture of embryo development by mapping the spatial organization of genes in cells of the embryo, revealing the relationships between gene expression patterns and embryonic development. For example, sagittal sections from mouse embryos spanning E9.5 to E16.5 with one-day intervals have been spatially mapped to achieve a global view of organogenesis at cellular resolution[7]. Similarly, a two-dimensional (2D) section of a whole E10 embryo was also spatially charted, defining the anatomic annotation of major tissue regions[6]. However, the cells in the embryo exist in a three-dimensional environment where structure, morphology, and characteristic biophysical and biomechanical signals play a significant role in influencing cell functions, such as migration, proliferation, and interaction, as well as patterning and axes formation. It is worth noting that some cellular processes that govern differentiation and morphogenesis tend to occur more efficiently in three dimensions rather than two dimensions. Therefore, it is crucial to profile the spatial transcriptome of tissue and organ-specific microarchitecture by aligning serial tissue sections of mouse organogenesis embryos to a 3D template, which has not yet been reported.

Embryo organogenesis, characterized by synchronized cellular and morphological changes, lays the foundation for the functional manifestation of organs. The early stages of organogenesis, specifically embryonic day 9.5–13.5 (E9.5–E13.5), remain particularly intriguing due to its immediate implication for organoids studies, tissue engineering and dissecting major developmental defects[2,9]. Moreover, understanding the co-evolution and 3D interactions of various organs will be instrumental in designing the strategies for in vitro organogenesis[10–12]. Here, utilizing the 10× Visium platform which provides high gene-detecting ability and substantial cell coverage, we built an organogenesis spatial atlas that encompasses collective transverse sections of the E13.5 mouse embryos. This spatial atlas of organogenesis represents almost all organ primordia configured at this stage and offers insight into the dynamic cell location, cell–cell communication, spatial heterogeneity, and organ architecture formation at the whole-embryo scale, consequently providing a molecular basis for understanding cellular interactions and allocation during mouse organogenesis.

## Results

### Construction of a spatial transcriptomic atlas of embryo organogenesis at E13.5

To generate a comprehensive molecular architecture of embryo development at E13.5, we utilized the 10× Genomics Visium platform to perform spatial transcriptomics sequencing on three individual mouse embryos. The whole embryo was serially cryo-sectioned into ~1000 sections at a thickness of 10 μm along the craniocaudal axis. To create a comprehensive and concise representation of the anatomic structures of E13.5 mouse embryo, we collected a total of 10 sections from a male embryo (Embryo 1, E1) that were spaced approximately every 100 sections apart, serving as a reference three-dimensional atlas (Fig. 1a). RNA-sequencing of these sections generated a high quality of spatial molecular map, with a median depth of 244.7 million reads per library, 16,418 spots from the entire sections, a median of 5668 genes and 22,253 unique molecular identifiers (UMIs) per spot (Supplementary Fig. 1a, b). We compared two sections from a replicate embryo (E2) and found that gene expression correlation, cluster identifications and spatial expression of marker genes are consistent, suggesting a high reproducibility of the spatial transcriptome and spatial expression domains among different embryos (Supplementary Fig. 1c–o). Meanwhile, the spatial distribution of marker genes also matched with in situ hybridization (ISH) data obtained from Mouse Genome Informatics (MGI) or Allen Brain Atlas (ABA)[13] (Supplementary Fig. 1o). We then focused our analysis on tissue sections from the reference embryo.

To systematically reveal the spatial molecular architecture of the E13.5 mouse embryo, we merged all 10 sections of the embryo and performed dimensional reduction using principal component analysis (PCA) and unsupervised clustering. By applying the uniform manifold approximation and projection (UMAP)[14], the spatial clusters were separated into two major groups: head and body parts (Fig. 1b). We identified 19 consensus spot clusters and annotated them based on the expression of the signature genes, enriched Gene Ontology (GO) terms and tissue images (Fig. 1b–d, Supplementary Fig. 2a, b and Supplementary Data 1). The spatial transcriptome atlas covered the majority of tissue and organs at this stage, including the brain cerebrum, spinal cord, respiration tract systems, gastrointestinal tract systems, circulation system, bone, skin, gonad, etc., providing a comprehensive view of the organs from head to tail. We found that the spatial clusters aligned well with the defined spatial anatomy structures when mapping spots back to the original coordinates of the tissue sections (Fig. 1c and Supplementary Fig. 2b). The D1-muscle domain (marked by *Myog* and *Tnnc2*, Fig. 1e and Supplementary Fig. 2a, b) were consistently observed across all seven sections of the body parts, while brain-related clusters such as domain D7-midbrain, D11-telencephalon, D14-neopallium, and D12-cephalic mesenchyme, as well as blood vessels (marked by *Ptgds* and *Atp1a2*, Fig. 1e and Supplementary Fig. 2a, b), were predominantly distributed in the head parts. Meanwhile, spatial domains D16-medulla oblongata and spinal cord, D13-ganglion (marked by *Gal* and *Sncg*, Fig. 1e and Supplementary Fig. 2a, b), and D9-cartilage matrix were consistently presented across the assayed embryo sections due to their ubiquitous existence (Fig. 1b, c and Supplementary Fig. 2b, c). In addition, we also identified organ-specific clusters. For example, D6-hepatic parenchyma, characterized by specific expression of *Afp* and *Apoa2*[15], was predominantly located in the liver; D19-heart exhibited differential expression of *Nppa* and *Myh6*, which are associated with heart contraction and blood circulation regulation (Fig. 1d and Supplementary Fig. 2a, b)[16,17]. Furthermore, to evaluate the spatial expression of identified top marker genes, we performed whole mount in situ (WISH) and verified the specific expression of *Sfn*, a relatively new marker gene in D15-skin, and *Pantr1*, which showed specific expression in D14-neopallium (Supplementary Fig. 2d–f). Altogether, this organism-level of spatial atlas provided a holistic molecular annotation for tissue architecture of mouse organogenesis at E13.5.

In addition, as the sex of the embryo at E13.5 has been determined, we collected four additional sections from the body part of a female embryo (E3) to accommodate a better coverage of sex differentiation (Supplementary Fig. 3). The sex of these embryos was confirmed by PCR and the spatial expression of marker genes (Supplementary Fig. 3c, d). By merging all sections from these two embryos (E1 and E3) together, we observed a minimum batch effect (Supplementary Fig. 3e), indicating high compatibility of the spatial transcriptome between these two embryos. Consequently, we annotated the spatial region of E3 by transferring labels from the represented embryo E1. The annotation of the spatial domains aligned consistently with the spatial anatomic structure and matched well with those sections from comparable positions in E1 (Supplementary Fig. 3f–i).

Next, we further evaluated the dataset of 10 sections from embryo E1 by integrating it with E13.5 data from the MOSTA dataset, which was collected on one sagittal section using stereo-seq technology[7]. Despite the use of different spatial technologies, these two datasets showed a compatible pattern as visualized in the UMAP (Supplementary Fig. 4a, b). We observed relatively consistent labeling of spatial regions by transferring annotations of our dataset to the MOSTA dataset[7] (Supplementary Fig. 4c–e). Taken together, our dataset of 10 sections illustrated a not complete but reasonable representation of the major organs and tissues of the embryo, spanning from head to tail, with minimal sampling bias. Therefore, it has the potential to serve as a core spatial atlas for E13.5 mouse embryos.

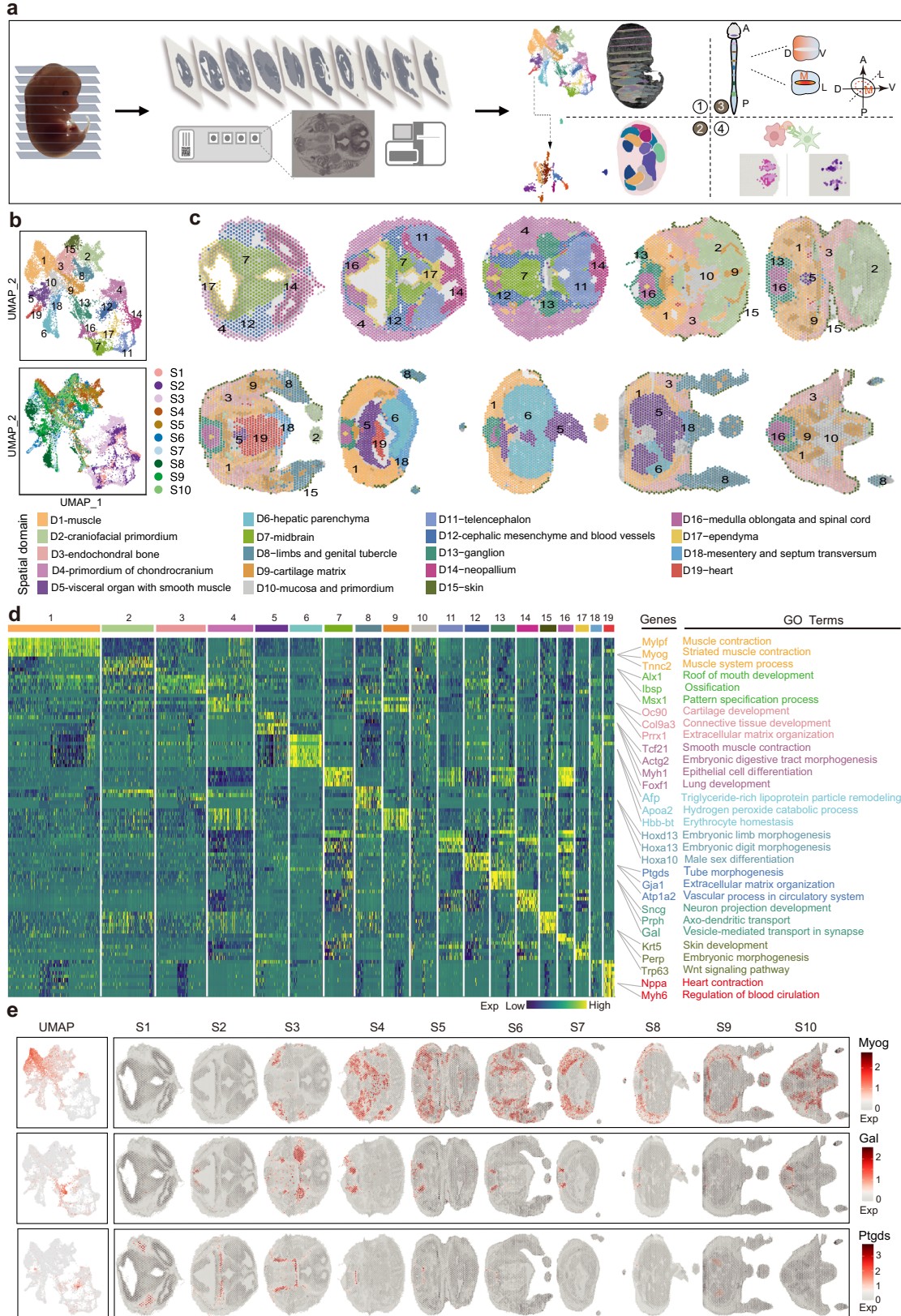

**Fig. 1 | 3D spatial transcriptional atlas for mouse organogenesis at E13.5.**
**a** Schematic overview of experimental design and analysis workflow for the spatial transcriptome of mouse organogenesis at E13.5. **b** UMAP projection and clustering of the spatial transcriptome from all sections and spots, colored by spatial domains (Top) and sections (Bottom). **c** The spatial distribution of 19 UMAP clusters in (**b**) across all embryo tissue sections, annotated according to anatomical structures

and molecular features in (**d**). **d** The heatmap showing the expression pattern of the top five representative marker genes for each spatial domain. Examples of marker genes are listed together with the enriched Gene Ontology (GO) terms for selected spatial domains. **e** UMAP projection and spatial expression distribution of selected marker genes, *Myog*, *Gal*, and *Ptgds* across all the ten embryo sections. S section, D domain.

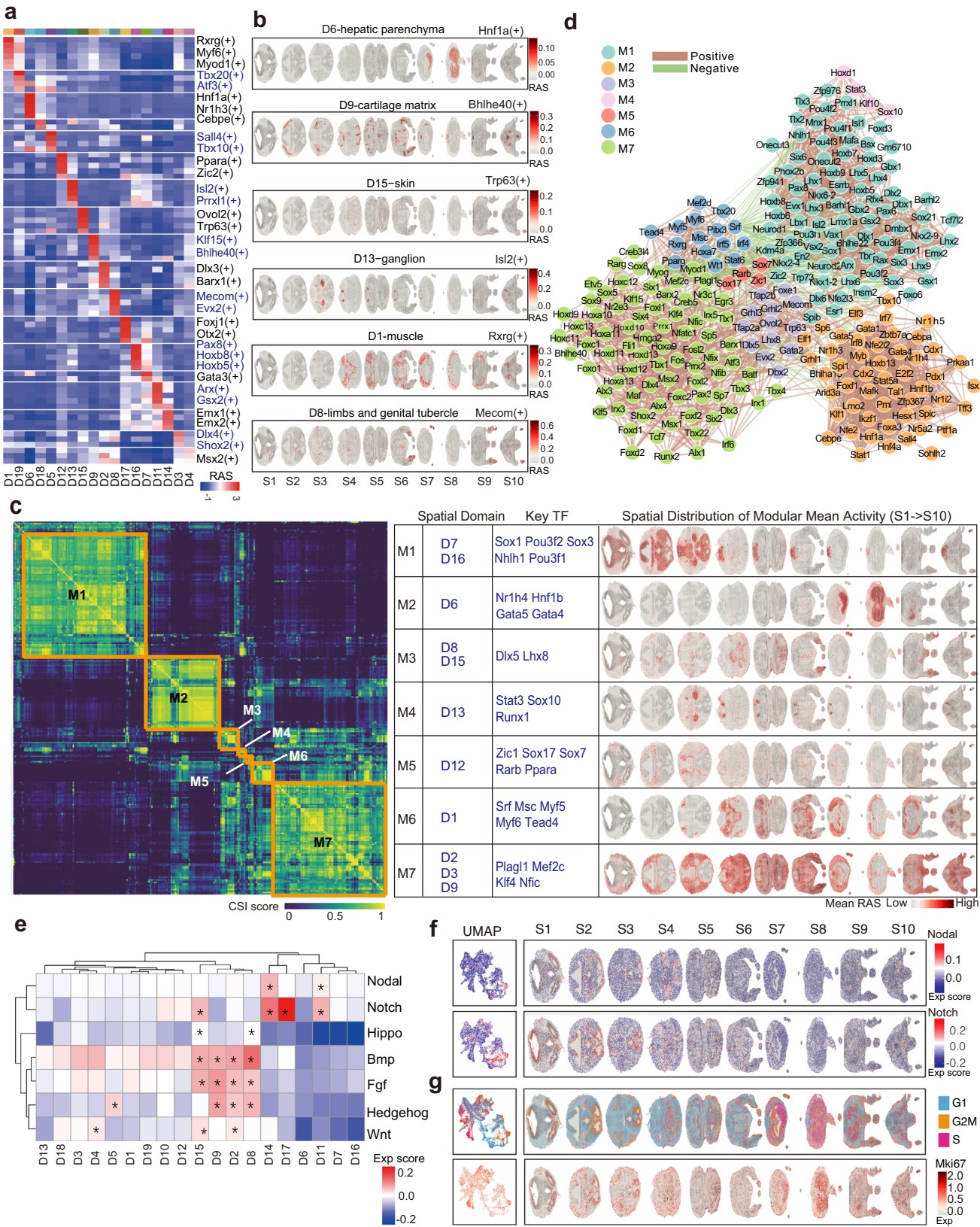

To facilitate the intuitive visualization of the spatial characteristics, we reconstructed a 3D embryonic model and depicted the spatial view of the identified spatial domains within the virtual embryo model (see the "Methods" section and Supplementary Movie 1). We also developed a web portal that provides a 3D illustration of spatial gene expression (http://most.ccla.ac.cn). These efforts highlighted the 3D recapitulation of molecular activities in the organogenesis embryo through meticulous alignment and collective integration of multiple spatial-transcriptome (ST) sections.

**Fig. 2 | Spatial gene regulation network, signaling and proliferation activity.**
**a** Heatmap of mean regulon activity score of selected top regional specific regulons for each spatial domain. **b** Spatial distribution of regulon activity scores across embryonic tissues for selected domain-specific regulons. **c** The hierarchical clustering of heatmap showing the seven regulon groups based on CSI matrix, with associated spatial domains, representative TFs and spatial plots visualizing the spatial distribution of mean activity score across each section. **d** The co-expression network based on CSI for the seven regulon groups. The node with different colors represents the regulon in each module and width of edge represents the CSI value of two nodes (filtered with CSI > 0.85). Color of the edges represents positive (brown) or negative (green) correlation. **e** Activity of development-related signaling pathways in each spatial domain. Differentially activated signaling in spatial domains computed by two-sided Wilcoxon rank-sum test are marked with * for mean score greater than 0 and p-value less than 0.001. **f** UMAP and spatial plots showing activities of Notch and Nodal signaling in all spots and all sections of embryo tissue. **g**. Spatial plots showing the spatial distribution of cell cycle activity of G1, G2M and S phase in embryo tissue and the spatial expression of *Mki67*.

## Regionalization and orchestration of gene regulatory network activity for organ development

To reveal the regionalized transcriptional regulatory activity underlying spatial gene expression patterns, we implemented the single-cell regulatory network inference and clustering (SCENIC) pipeline[18] and calculated the regulon activity score (RAS) for each spatial spot. We found the spatial domains based on RAS were consistent with the spatial gene expression clusters (Supplementary Fig. 5a–c), indicating that the activity of transcription factors (TFs) regulatory network may be involved in determining the cell fates and locations.

We then sought to systematically identify critical TF regulators associated with each spatial domain by computing the regulon specificity score (RSS) of each regulon for all 19 spatial domains based on the Jensen–Shannon divergence[19]. We identified significantly enriched regulons in each spatial domain (Fig. 2a, b and Supplementary Data 2 and 3). This allowed us to obtain organism-level key TF networks that specifically function in a location- and cell-type-dependent manner (Fig. 2a, b and Supplementary Fig. 5d). For example, Hnf1a, Nr1h3 and Cebpe regulons were identified as top regulons in D6-hepatic parenchyma, in agreement with their known functions in hepatocyte development, differentiation or bile acid homeostasis[20–23]. The Bhlhe40 regulon showed specific and strong activity in the D9-cartilage domain, while Trp63 exhibited specific activity in the D15-skin domain. Similarly, in pan-muscle cells, Myf6 and Myod1 were identified as master TF regulons[24–26]. Notably, Isl2 and Prrxl1 emerged as top-ranked regulons in the D13-ganglion region, and Dlx3 regulon exhibited high specificity score in the domain of D2-craniofacial primordium, consistent with their reported roles in the regulation of craniofacial bones development[27]. Of note, besides known cell type markers, the 3D spatial atlas delineated unappreciated TFs enriched in each respective organ (Supplementary Data 3), such as Rxrg in muscle cells and Mecom in limb tissues (Fig. 2a, b)[28]. We also observed specific expression of TFs in enriched spatial domains, which was relatively restricted compared to the RAS of the corresponding regulons (Supplementary Fig. 5d, e). These results suggest that the unrecognized spatial-domain-specific regulons identified from our spatial atlas could serve as an important resource for further functional analyses.

During embryogenesis, TFs often coordinate and regulate gene expression in a combinatorial manner. Understanding the divergent or convergent TF regulation mechanisms, both within and outside developmental lineages, is of particular interesting. To further reveal the regionalized transcriptional regulation networks in our spatial atlas, we calculated the similarity of top selected regulons using the connection specificity index (CSI)[29]. Hierarchical clustering was then applied to identify potential function-related regulon groups, by computing the averaged RAS score for each TF regulon module across spots. We discovered distinct spatial regulon modules that highlighted the orchestration of TF regulons in particular tissue regions (Fig. 2c). Interestingly, the TF regulons formed seven co-activation modules for the 19 spatial tissue domains, indicating a shared co-regulatory mechanism for particular organs during this early stage of organogenesis. For example, regulon module 1 (M1), which includes key regulators such as Sox1 and Pou3f2, showed high activity in D7-midbrain, D16-spinal cord, and cerebrum domains, indicating a general regulation mechanism for neural development. Regulon module 7

(M7) was associated with neural crest derivatives, such as craniofacial, endochondral bone and cartilage development. To reveal the underlying connections in each module, we further constructed a TF co-expression network on the basis of their CSI value (Fig. 2d). Collectively, this network analysis of regulatory interactions provided insights into the spatially dissected regulatory mechanisms across different cell types and locations in controlling embryo organogenesis.

## Spatial signaling pathway activity at the whole embryo scale

The patterning and regionalization of embryos during development are highly dependent on morphogenetic signals. To systematically map the spatial distribution of signaling pathways activities in the entire embryo, we examined the enrichment score of signaling genes (including ligands, receptors, key signaling effectors and regulators) for seven pathways: Wnt, Bmp, Fgf, Hippo, Nodal, Notch and Hedgehog. We gained a comprehensive view of spatial signaling landscape at both global and individual spatial-domain level (Fig. 2e, f).

Of note, the spatial distribution of the developmental signaling scores revealed relatively high activity across the telencephalon, ependyma, limb, cartilage and skin, while showing particularly low activity in the liver, midbrain and spinal cord domains. This suggests differential induction and patterning scheme in different tissues and organs. For example, Nodal signaling mainly exhibited strong activity in regions of the cerebrum, part of the midbrain and jaw area of the craniofacial primordium, indicating its important roles in neurogenesis and specifying the function of the mouth[30] (Fig. 2e, f). On the other hand, Notch signaling, which mediates juxtracrine cell-cell communication[31,32], showed enrichment in the brain and spinal cord ependyma region, cerebrum, and part of the skin domain. Bmp signaling activity was relatively high in the limb and less represented in liver and cerebrum and midbrain domain, consistent with previous studies[33,34] (Fig. 2e). Hedgehog and Fgf signaling also showed moderate activity in the cartilage and limb, suggesting that these pathways may coordinate with Bmp to play important roles in limb and bone morphogenesis (Fig. 2e).

Furthermore, we assessed the spatial distribution of the proliferation state by calculating the "Cell cycle score". Regions containing nerve progenitors, such as D14-cerebrum, part of D11-cerebrum and D17-ependyma, exhibited high levels of G2M and S phase score. In contrast, the midbrain and the ganglion domains showed relatively low activity scores for both G2M and S phases (Fig. 2g). Hence, this analysis uncovered the regionalization of proliferation activity at the whole-embryo scale.

## Spatially resolved molecular characterization of major organs

In the organism-level classification, the spatial domain of D5 (Fig. 1c) was a major cluster consisting of spots from multiple visceral organs. To achieve a finer annotation, spots of D5 were subjected to further unsupervised clustering and we obtained 10 sub-clusters (Fig. 3a, b), each with distinct spatial expression patterns, anatomical structure, specific expressed marker genes and enriched GO terms. We assigned these sub-clusters as pancreas, bladder, gonad, stomach, gut, metanephros, lung, umbilical, trachea and mesenchyme (Trachea_Mes) and mesonephros-spleen-superior recess of omental bursa (Nep_sp_Om). This finer annotation provided a full spectrum of all major visceral

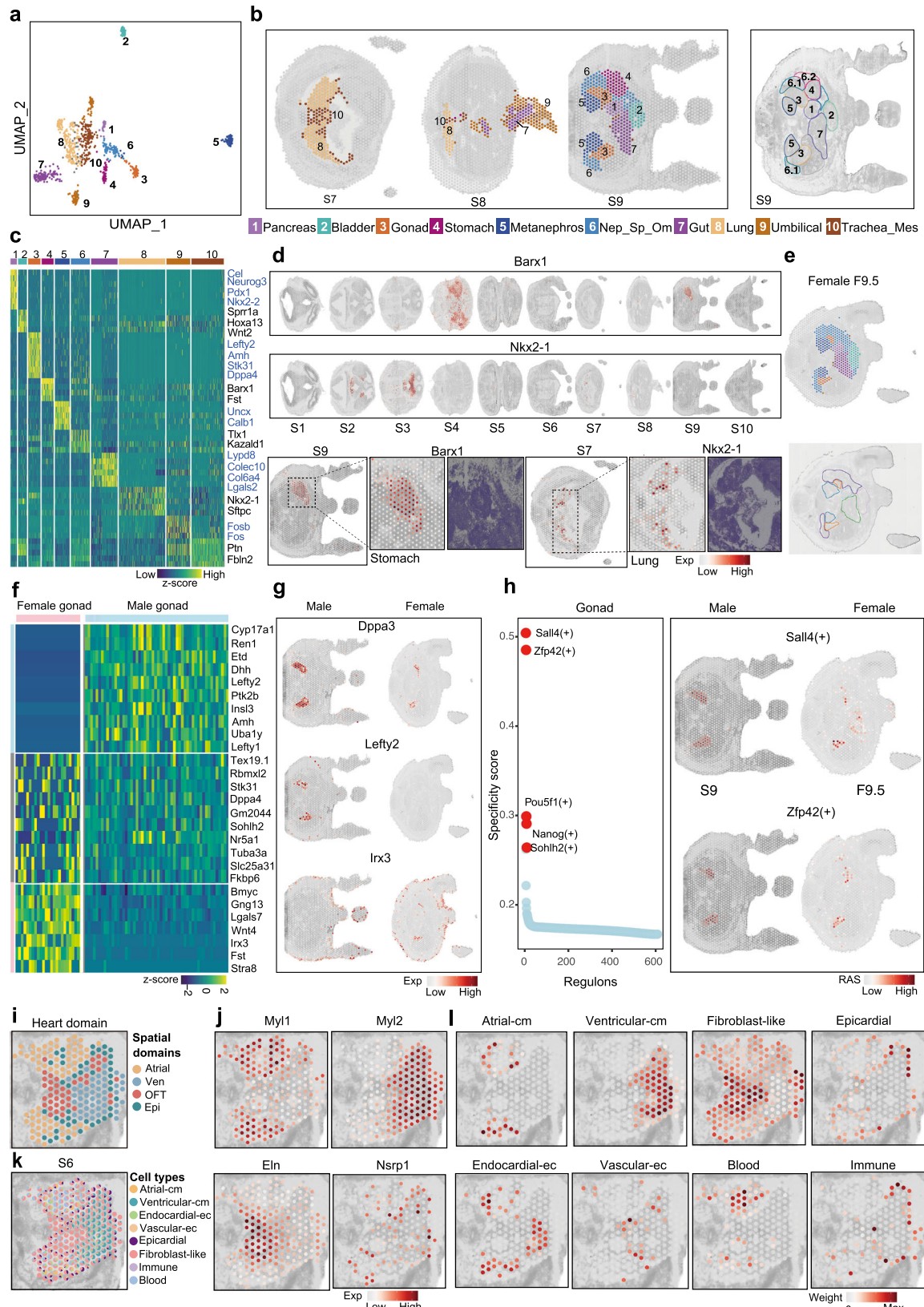

organs during the mid-organogenesis embryo (Fig. 3b, c and Supplementary Fig. 6a–c), which allowed us to define visceral organ-specific signatures (Supplementary Data 4). For example, we found *Nkx2-1*, a key factor known for its role in regulating the development of brain and lung structures, showed specific expression in both lung and brain regions[35] (Fig. 3d). The stomach domain was characterized by the

expression of tissue-specific gene *Barx1* (Fig. 3d), which has been shown in the control of thoracic foregut specification and tracheo-esophageal septation[36]. Additionally, we identified less-studied genes such as *Fst*, *Tmem200a* and *Egln3* co-expressed with *Barx1* in the stomach domain, indicating their potential involvement in stomach development (Supplementary Fig. 6b).

**Fig. 3 | Spatially resolved molecular characterization of major organs and sex specification. a** UMAP embedding of spots from D5 (visceral organ) labeled by subclusters in embryo E1. **b** Spatial distribution of the 10 subclusters of D5. Segmented regions are highlighted on S9 (right) and colored according to the denoted anatomical structure. Subcluster 6 Nep_sp_Om was further divided into 6.1 and 6.2 to represent finer structures. Trachea_Mes, trachea and mesenchyme; Nep_sp_Om, mesonephros-spleen-superior recess of omental bursa. **c** The heatmap showing the expression pattern of selected top marker genes for each subcluster. **d** Spatial expression of selected marker genes *Barx1* for Stomach (top) and *Nkx2-1* for lung (middle) across all embryo sections. The dashed box showing the region-specific expression of *Barx1* in the stomach region of S9 (bottom left) and *Nkx2-1* (bottom right) in the lung region of S7 and their corresponding regions in tissue images. **e** Subcluster annotation of D5 (top) and segmented regions are highlighted according to the denoted anatomical structure (bottom) on section 9.5 (F9.5) in embryo E3. **f** Heatmap showing the sex-related gonad-specific marker genes for the female and male embryos, of which pink represents female-specific, light blue represents male-specific, and gray represents common-specific marker genes. **g** Spatial distribution of sex-related gonad-specific marker genes *Dppa3* for common, *Lefty2* for male and *Irx3* for female in S9 of male embryo E1 and F9.5 of female embryo E3. **h** Rank plot for regulons in gonad based on regulon specificity score (left). Spatial distribution of regulon activity scores for regulon Sall4 and Zfp42 in both male and female tissue sections. **i** The spatial distribution of annotated subclusters of D19-heart. Ven, Ventricular; OFT, Outflow tract; Epi, Epicardial. **j** Spatial expression of selected sub-domain specific marker genes *Myl1* (Atrial), *Myl2* (Ven), *Eln* (OFT), and *Nsrp1* (Epi). **k** The spatial map of predicted cell types in the heart region. **l** Spatial visualization of deconvoluted weights of 8 heart-specific cell types including atrial cardiomyocytes (atrial_cm), ventricular cardiomyocytes (ventricular_cm), endocardial endothelial cells (endocardial_ec), vascular endothelial cells (vascular_ec), epicardial cells, fibroblast-like cells, immune cells, and blood cells. S section, D domain.

With its high sensitivity and comprehensive tissue coverage, the spatial atlas allowed us to investigate signature genes of embryonic visceral organs that have not been adequately explored. For example, *Sprr1a* was found to have spatially exclusive expression in the bladder (Supplementary Fig. 6b), while *Uncx*, *Calb1* and *Foxd1* showed spatially restricted expression in the metanephros regions. *Lypd8, Colec10* and *Col6a4* were specifically expressed in the gut. Among these subdomains, the pancreas expressed more region-specific genes, indicating its strong and unique organ characteristics (Supplementary Fig. 6b). *Cel, Neurog3, Nkx2-2*, and *Pdx1* were identified as markers for pancreas development.

Since the represented embryo (E1) was male and the collected sections of the repeated female embryo (E3) also covered the regions of the visceral organs, we annotated and identified the spatial locations of all subclusters except for the Stomach in female visceral organs by applying for label transfer from E1 (Fig. 3e). The specific expression of the top marker genes in gut, metanephros, bladder and pancreas were further validated in the corresponding section from the female embryo (Supplementary Fig. 6d). For the development of Gonad, which involved sex determination and differentiation at E13.5, the spatial expression of common and sex specific gonad genes were explored (Fig. 3f, g). Interestingly, *Lefty1* and *Lefty2* genes were found to be specifically expressed in the male gonad, which have been reported to regulate the male germ cell fate segmentation[37]. Several genes such as *Irx3*[38] and *Lgals7*, were identified specifically expressed in female gonad. A set of gonad-specific genes in both female and male, such as *Dppa3* and *Text19.1*, was also identified (Fig. 3g and Supplementary Fig. 7a). Functional enrichment analysis showed that female and male-specific genes were highly related to the corresponding sex development, while the common gonad-specific genes were involved in DNA modification, such as alkylation or methylation, besides the terms related to gonad development, indicating the critical role of dynamic DNA modification involved in gonad differentiation (Supplementary Fig. 7b).

Furthermore, we also identified regional-specific TF regulons that distinguish the subregions of these visceral organs (Supplementary Data 5, Fig. 3h and Supplementary Fig. 7c–e). For example, Neurog3, Nr5a2 and Ptf1a were associated with pancreas development[39, 40] (Supplementary Fig. 7c), while Cdx2 and Nkx2-1 were the key regulons for gut and lung, respectively (Supplementary Fig. 7d, e). Sall4 and Zfp42 were found to be highly expressed regulons associated with Gonad (Fig. 3h)[41]. The regulon activity score of these organ-specific regulons was examined and validated in the repeated embryo E3 (Fig. 3h and Supplementary Fig. 7d, e), showing a consistent activation pattern in both male and female embryos. These findings thus characterized previously unappreciated tissue structures at the transcriptional level within the subclusters of visceral organs.

Next, we sought to illustrate the fined and detailed molecular structures in the sub-organ resolution during the organogenesis of mouse embryos. Using the developing heart as an example, we identified four spatial molecular sub-regions by unsupervised clustering of spots extracted from D19-heart of section 6 (S6) of E1. These spatial domains matched well with typical heart anatomical structures, and we annotated them as Atrial, Ventricular (Ven), Outflow Tract (OFT), Epicardium (Epi) respectively (Fig. 3i). Subsequently, we identified a set of transcriptional features specific to each sub-region of the heart at this developmental stage (Supplementary Fig. 8a, b and Supplementary Data 6). For example, *Myl1* and *Sln*, which are well-known Atrial maker genes, showed restricted expression in the Atrial domain. *Msln* and *Nsrp1* displayed epicardium-specific expression, in which *Msln* was previously reported to be specifically expressed in the epicardium[42], and *Nsrp1* is relatively new (Fig. 3j). Furthermore, we assessed and illustrated the spatial distribution of cell types involved in heart development by performing deconvolution analysis using corresponding heart scRNA-seq data[43] (Fig. 3k, l), and the cell type distribution also agrees with the anatomical structure. Finally, we investigated the spatial expression pattern of genes associated with congenital heart diseases (CHD) and identified a group of CHD-associated genes that share a similar spatial pattern and show high expression in the OFT sub-region at this mid-organogenesis stage (Supplementary Fig. 8c–e). Of note, *Eln*, a gene associated with pulmonary valve atresia (PA)[44], and *Mgp*, which is related to peripheral arterial disease, were included in this cluster. Collagen-related genes *Col3a1* and *Col1a2*, which were not specifically associated with a certain type of CHD, also showed high expression in the OFT region. Although limited to one stage, these findings shed light on potential mechanisms underlying CHDs. Reassuringly, we further repeated cell type decomposition and validated the spatial expression of the top marker gene in the repeated female embryo E3 (Supplementary Fig. 8f–h). Altogether, these findings showcase the valuable utilities of this spatial atlas in dissecting sophisticated tissue organizations.

## The craniocaudal, dorsoventral and radial axes in establishing the spatial patterning of spinal cord

During embryonic development, a crucial event at around E13.5 is the acquisition of positional identities within the neural tube, which ultimately gives rise to the brain and spinal cord. The neural tube undergoes patterning along the craniocaudal, dorsoventral and radial axes to establish distinct domains. Previous studies utilizing scRNA-seq have provided many insights into nervous system development[45–47]. Herein, we aimed to explore the spatial patterning of the neural tube at the whole embryo scale.

One of the key regulators involved in establishing the regional identities along the anterior–posterior (AP, craniocaudal) axis of both the hindbrain and spinal cord is the family of Homobox (Hox) genes. These genes form a 'Hox code' and play a critical role in determining the boundaries and positions of different neuronal subtypes along the AP axis[48]. Hox genes are generally categorized

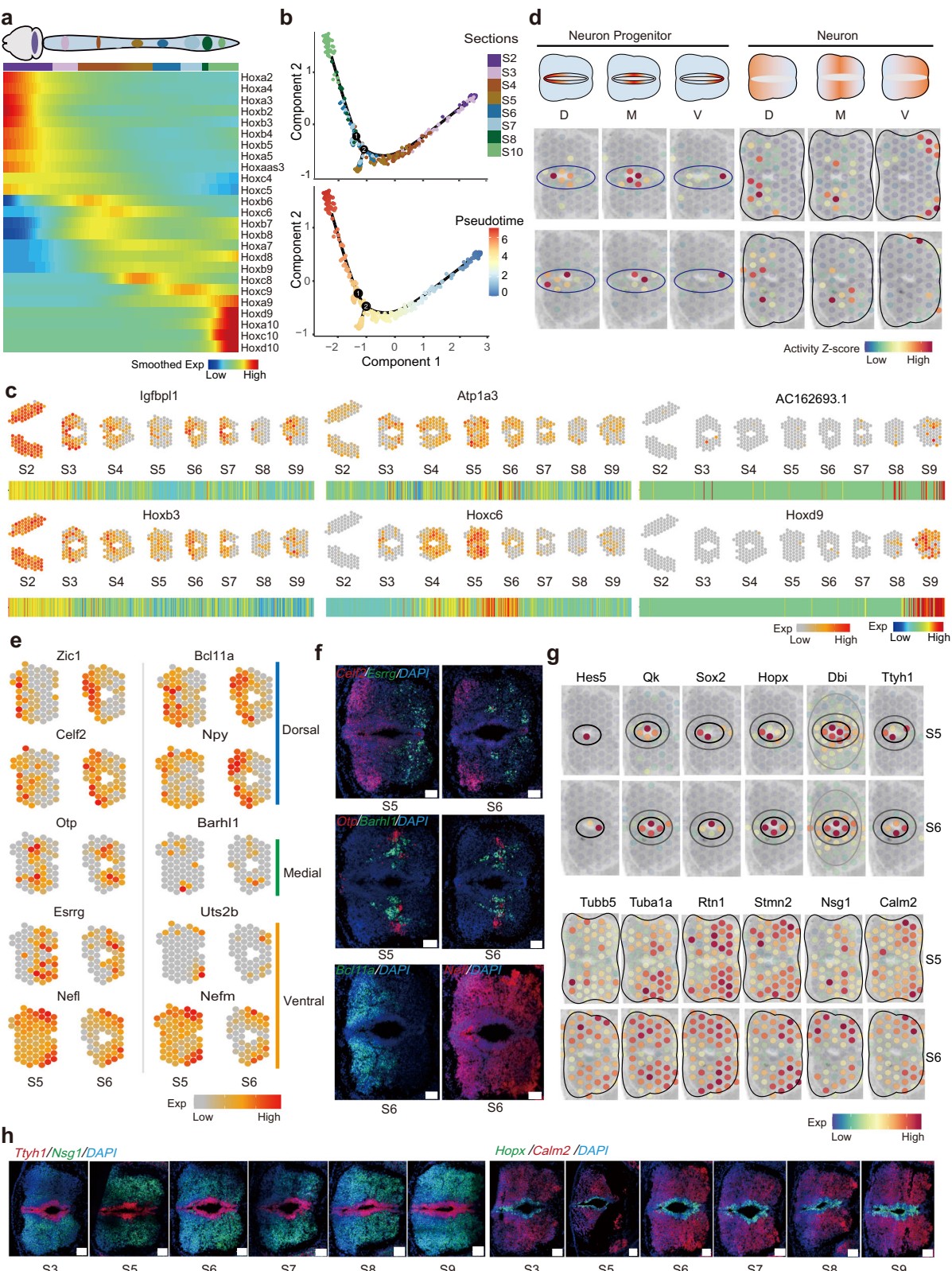

into anterior (Hox 1–3/4), trunk/central (Hox 4/5–9) and posterior (Hox 10–13) paralogues groups (PGs), reflecting their arrangements along each genome cluster[49]. We first examined the expression of a total of 26 expressed Hox genes in the spatial domain of D16-medulla oblongata and spinal cord (Fig. 4a, b). As expected, the anterior Hox PGs were predominantly expressed in the rostral sections, while the

posterior Hox PGs showed increased expression in the posterior regions (Fig. 4a). In order to identify new genes involved in A–P patterning, we arranged the spatial spots along the anterior to posterior plane ordered by the expressed Hox genes to establish a pseudo-space trajectory (Fig. 4b). Next, we performed differential gene analysis to identify genes that showed distinct A–P patterning

**Fig. 4 | 3D alignment to reveal body axes and spinal cord patterning. a** Heatmap plot showing the smoothed spatial expression pattern of Hox family genes along the A–P axis with spots from hindbrain and spinal cord regions ordered by pseudo-axis within each section. **b** Pseudo-time trajectory plot showing pseudo-space patterning of spots from hindbrain and spinal cord region across sections of whole mouse embryos, colored by section numbers (top) and pseudo-time (bottom). **c** Spatial expression of selected Hox family genes and newly identified A–P axis related genes in hindbrain and spinal cord from sections along anterior to posterior (top), and heatmap showing the expression pattern of corresponding genes ordered along sections from anterior to posterior (bottom). **d** Spatial plot showing

the D–M–V activity scores for respective neuronal progenitors and neuronal cells in the spinal cord of sections 5 and 6. **e** Spatial expression of selected D–V axis-related genes in sections 5 and 6. **f.** RNAScope multiplex in situ hybridizations of D–V patterning genes in the spinal cord, and representative images from hybridizations on sections 5 and 6 ($n = 3$). Scale bars, 100 μm. **g** Spatial visualization of radial axis patterning genes in the spinal cord region. **h** RNAScope multiplex in situ hybridizations of radial axis patterning genes in the spinal cord, and representative images from hybridizations on serial sections from anterior to posterior ($n_{s5,6,7} = 3$, $n_{s3,8,9} = 2$). Scale bars, 100 μm. Source data are provided as a Source Data file.

along this pseudo-space trajectory. To further refine these candidate genes, we applied correlation analysis between the gene expression of each spot and the combinatorial pattern of AP slices (Supplementary Fig. 9a and Supplementary Data 7). We identified A–P region-specific genes including *Igfbpl1* (Fig. 4c) in the rostral region, *Atp1a3* in the central and *AC162693.1* in the relative caudal region (Fig. 4c). These newly identified genes expand our understanding of potential regulators involved in establishing the body plan.

We also examined the dorsal–ventral (D–V) patterning during embryonic spinal cord development, taking advantage of our transverse collection strategy. Along the D–V axis, diverse neurons with distinct structures and functions are generated, controlled by precise spatial expression genes and signals[50,51]. To investigate this patterning, we allocated the combinatorial expression of a curated marker gene list that is used to define different domains of progenitors in single-cell RNA-seq[45] to the spinal cord spots, including dorsal gene sets (D) by combinatorial marker expression of roof plate (RP) and dp1–dp6, medial gene sets (M) by marker territory of p0–p2, as well as ventral (V) gene sets by pMN, p3 and floor plate (FP) genes. Similarly, we categorized gene sets for neuronal regions into dorsal (D, marked by dl1–dl6), medial (M, marked by V0–V2b) and ventral (V, marked by Mn and V3) subdomains. This allowed us to observe clear separation of the dorsal, medial and ventral structures within the spinal cord (Fig. 4d). To predict new genes involved in spinal cord patterning, we identified domain-specific genes along the D–V axis in different spinal cord regions (Fig. 4e, Supplementary Fig. 9b and Supplementary Data 8). For example, *Zic1*, *Bcl11a*, *Celf2* and *Npy* were specifically expressed in the dorsal region, while *Otp* and *Barhl1* were up-regulated in the medial region. Furthermore, genes such as *Esrrg*, *Uts2b*, *Nefl* and *Nefm* were highly expressed in the ventral region (Fig. 4e). We performed single molecular fluorescence in situ hybridization (smFISH) on serial sections of the spinal cord (Fig. 4f and Supplementary Fig. 9c) and validate the expression of D–V patterning-related genes. Accordingly, we further identified TF regulons associated with both A–P and D–V patterning (Supplementary Fig. 10a–d). As expected, the Hox TF family ranked highly in spatial patterning. Several Sox TFs, known for their importance in neural development[52], were also identified as regulons associated with A–P patterning. Notably, Mnx1 exhibited specific regulon activity in the ventral part, consistent with its role as a TF specific to spinal cord motor neurons[53].

To further investigate the spatial differences along the radial axis (medial–lateral) that are relevant to neural progenitor proliferation and differentiation, we conducted differentially expressed gene (DEG) analysis along the inner and outer layers of spinal cord spots (Fig. 4g). We found well-known marker genes of neural progenitors such as *Hes5*, *Qk* and *Sox2*[54–57], as well as genes with limited study in neural progenitor cells, such as *Hopx*, *Dbi* and *Ttyh1* which were expressed in the inner regions. Whereas genes including *Tubb5*, *Tuba1a*, *Rtn1*, *Stmn2*, *Nsg1* and *Calm2*, which may be related to differentiated neuronal cells, showed high expression in the outer layers. We performed smFISH on serial sections from anterior to posterior, confirming the distinct patterns of expression for *Ttyh1*, *Hopx*, *Nsg1* and *Calm2* (Fig. 4h and Supplementary Fig. 9c). Our analysis therefore suggests that numerous additional genes with spatially restricted expression

patterns can be identified to constitute the regionalized patterning of neural tube. Hence the spatial atlas may provide new clues for neuronal specification and patterning of the developing spinal cord.

## Integration of single-cell and spatial atlases illustrates spatially resolved cell interactions

Each spot in the Visium platform is expected to consist of a mixture of around 20 cells and probably includes multiple cell types. To investigate the cell type heterogeneity in the spatial regions, we performed cell type deconvolution using Robust Cell Type Decomposition (RCTD)[58] with E13.5 mouse single cells derived from the TOME dataset as the reference[1].

We defined specific spatial distribution for 46 cell types within each spatial domain at the E13.5 stage (Fig. 5a, Supplementary Figs. 11–13 and Supplementary Data 9). As expected, myocytes were the most abundant throughout spatial regions, and endothelial cells and white blood cells were also widely distributed (Fig. 5a and Supplementary Fig. 11a). Visualization of the deconvoluted cell type weights on tissues showed that the assigned coordinate of organ-specific cells corroborated known tissue functions (Supplementary Figs. 11–13). For example, epithelium cells were exclusively located in the otic, branchial arch, lung, pancreatic and renal (Fig. 5b). By examining the top abundant cell types in each spatial domain, we observed a consensus between cell types and their anatomical structures. For example, D6-hepatic parenchyma was dominated by hepatocytes and definitive erythroid lineage cells (Fig. 5b and Supplementary Fig. 13b). Similarly, the dominant cell types in D19-heart were cardiomyocytes and endothelium, which are essential functional units of the heart (Fig. 5a, b and Supplementary Fig. 13b). Interestingly, early chondrocytes, osteoblast progenitors A, osteoblast progenitors B and connective tissue progenitors were located in the physically distinct but proximal areas (Supplementary Fig. 11b), indicating their potentially close interactions. Of note, posterior floor plate cells were detected within a limited area near the ventral region of the spinal cord, with only one to two spots per section across the body trunk, indicative of high fidelity of spatial mapping (Supplementary Fig. 12b). The identified marker genes in TOME dataset, such as *Slit1*, *Ntn1* and *Shh* shared a high expression level in the corresponding spatial regions of posterior floor plate[1] (Supplementary Fig. 12b).

To investigate how spatial proximity of cell types may influence each other and shape the signaling landscape to coordinate the developmental programs, we developed a spatial cell–cell communication (CCC) analysis workflow (named STcomm). STcomm integrates spatial cellular colocalizations with enriched ligand–receptor (L–R) co-expression patterns inferred from both spatial and single-cell transcriptomic data. The underlying assumption is that spatially co-localized cells within spot level can more reliably infer L–R mediated CCC (Fig. 5c and Methods). Firstly, we quantified the colocalization of cell-type pairs within spots by calculating the Pearson correlation coefficient (PCC) based on cell-type composition predicted by RCTD. We thus identified significant co-occurrent cell type groups that were present within the same spot, as revealed by the cell type colocalization network (Supplementary Fig. 14a). For example, Olfactory epithelium and Olfactory sensory neurons were

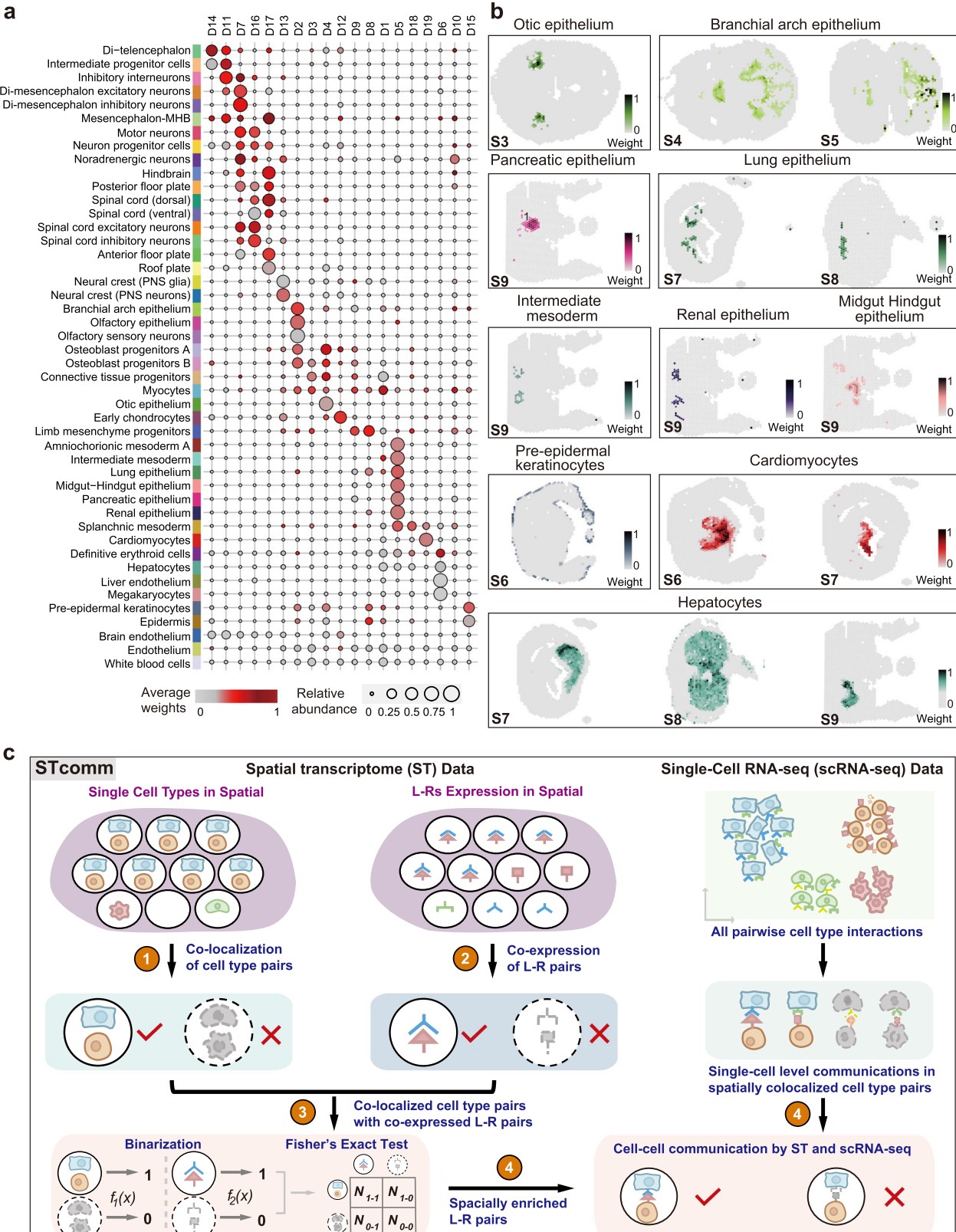

**Fig. 5 | Spatial mapping of cell populations across all the tissue sections.**
**a** Deconvolution analysis inferred the weights of 46 cell populations in all spots of embryo tissue. The dot plot showing cell type composition within each spatial domain. Color bar indicates the averaged cell-type weights in each spatial domain. Dot size represents the relative abundance of cell types in each spatial domain. **b** Spatial distribution of organ/tissue specific cells including epithelium,

intermediate mesoderm, pre-epidermal keratinocytes, cardiomyocytes and hepatocytes. **c** The workflow of STcomm analysis pipeline which combined the spatial cellular colocalization and L-R co-expression from spatial transcriptomic data with cell–cell communication inferred from sc-RNA seq data (see the "Methods" section). Source data are provided as a Source Data file.

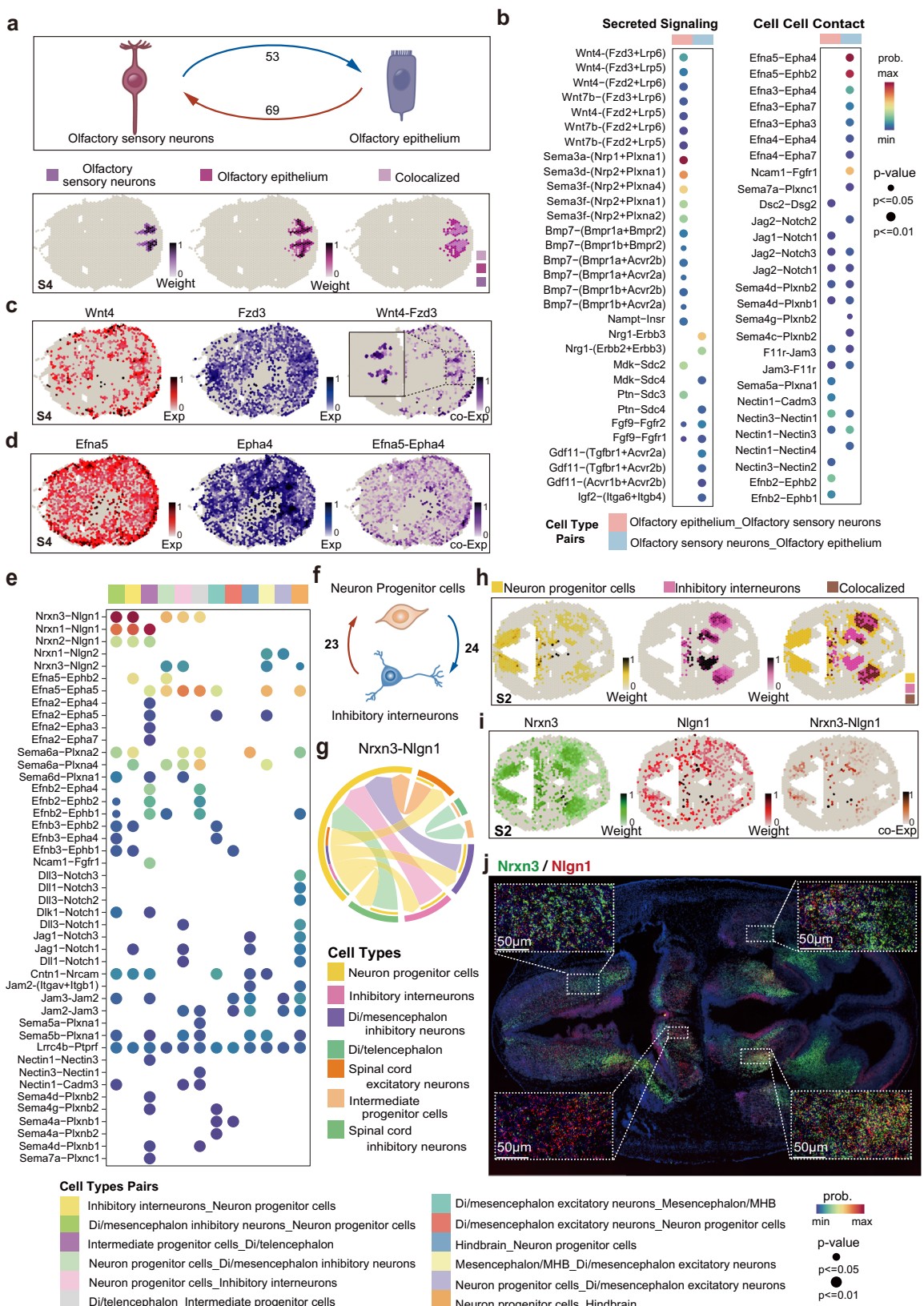

co-localized in craniofacial primordium at the front region of section 4 (Fig. 6a). Similarly, pre-epidermal keratinocytes and epidermis significantly co-existed and were primarily distributed in D15-skin, which agrees with the expected localization of these cell types (Supplementary Fig. 14a). Moreover, hepatocytes, definitive erythroid lineage, megakaryocytes and liver endothelium formed a

colocalization network, revealing the cellular composition within the liver microenvironment. Of note, we observed spatial proximity between neuron progenitor cells and inhibitory interneurons, hindbrain, Di-mesencephalan inhibitory neurons, Di-mesencephalan excitatory neurons, as well as excitatory and inhibitory neurons. Neural crest -PNS neurons and neural crest-PNS glia, which were

**Fig. 6 | Cell–cell communication network of spatial proximity brain-related cell types in mouse embryo organogenesis. a** Schematic showing the number of significant L–R pair interactions between olfactory sensory neurons and olfactory epithelium cells by STcomm analysis (top). The bottom panel showing the spatial mapping (color intensity) and colocalization of olfactory sensory neurons and olfactory epithelium cells according to deconvoluted weights in section 4 (S4). **b** Dot plot showing significant L–R pairs between olfactory sensory neurons and olfactory epithelium cells via secreted signaling (left) and cell-cell contact (right) with *p* < 0.05. The dot color represents communication probability and the size indicates p-values which are computed from one-sided permutation test by Cellchat. **c** The spatial distribution of expression of L–R pairs of *Wnt4* and *Fzd3*, and their co-expression in S4. The inlet shows the co-expression level of *Wnt4* and *Fzd3* in co-localized region of olfactory sensory neurons and olfactory epithelium. **d** The spatial distribution of expression of L–R pairs of *EfnaS* and *Epha4*, and their co-expression in S4. **e** Dot plot showing significant L–R pairs communication among

spatial proximity brain-related cell types calculated by STcomm. The dot color and size indicate communication probability and *p*-values which are computed from the one-sided permutation test by Cellchat. **f** Schematic showing the number of significant L–R interactions between neuron progenitor cells and inhibitory interneurons. **g** Circos plots representing significant interaction of L–R pairs of *Nrxn3* and *Nlgn1* among spatial proximity brain-related cell types. **h** Spatial plots showing the spatial distribution (color intensity) and colocalization with spots of neuron progenitor cells and inhibitory interneurons according to deconvoluted weights in section 2 (S2). **i** Spatial distribution of expression and co-expression of L–R *Nrxn3* and *Nlgn1* in S2. **j** *Nrxn3* and *Nlgn1* spatial expression pattern examined by RNA-scope in brain tissue sections matched to S2 in (**i**) (*n* = 4). White dashed box showing the staining of *Nrxn3* and *Nlgn1* in spatial proximity cells. Blue, DAPI; green, Nrxn3; red, Nlgn1; yellow, co-location of Nrxn3 and Nlgn1. Source data are provided as a Source Data file.

abundant in D13-ganglion domain (Fig. 5a), exhibited significant interactions with each other (Supplementary Fig. 14a).

Next, we performed Fisher's exact test[59] on binarized co-localized cell type pairs and co-expressed L–R pairs at spot level to identify significantly co-expressed L–R pairs for spatially co-localized cell types. We further calculated significant communication between L–R pairs within co-localized cell type pairs using reference single-cell transcriptomic data. At last, we retained spatially confident communications based on the above significance of Fisher's exact test (Fig. 5c and see the "Methods" section). Leveraging the spatial information, STcomm characterized confident cell-cell interaction within the tissue organization (Fig. 5c and Supplementary Data 10). Specifically, according to the identified spatial co-occurrence cell types, we examined the L–R interactions between olfactory epithelium and olfactory sensory neurons, which showed a high-frequency interaction. We identified 53 L–R pairs from olfactory sensory neurons to olfactory epithelium and 69 L–R pairs from olfactory epithelium to olfactory sensory neurons (Fig. 6a). Notably, in terms of secreted signaling molecules, the *Sema3* family, *Wnt* and *Bmp* originating from the olfactory epithelium, exhibited a high communication probability towards olfactory sensory neurons (Fig. 6b), consistent with their reported role in regulating axon outgrowth and navigation of olfactory sensory neuron[60–64]. Visualization of *Wnt4* and *Fzd3* expression on the tissues showed a high level of co-expression in the spots where olfactory epithelium and olfactory sensory neurons co-localized (Fig. 6c). Regarding cell–cell contact-mediated L–R interactions between these cell-type pairs, the *Efna-Epha* family and Notch signaling were identified to have a strong communication probability, indicating their important role in regulating the development of olfactory epithelium cells[65] (Fig. 6d).

E13.5 embryogenesis is hallmarked by rapid neurogenesis. We next focused on exploring the specific interactions among neuronal cells that contribute to this intricate composition (Supplementary Fig. 14a, b). We identified a set of L–R pairs with significant probability of communicating with each other among these co-localized neuron-related cells (Fig. 6e). Specifically, we discovered 23 L–R pairs originating from neuron progenitors and 24 L–R pairs originating from the inhibitory interneurons, which exhibited significant communication probability with each other (Fig. 6f). Among them, the *Nrxn3–Nlgn1* showed a prevailing communication probability between neuron progenitors and inhibitory interneurons (Fig. 6e, g and Supplementary Fig. 14c). To provide an example, we took the brain-containing section 2 (S2) and found that the spatial co-occurrence of the neuron progenitor cells and inhibitory interneurons aligned well with the spatial co-expression pattern of the *Nrxn3–Nlgn1* L–R pairs on the tissues (Fig. 6h, i). Similar results were observed in section 1 (Supplementary Fig. 14d). Furthermore, we validated the spatial proximity of *Nrxn3–Nlgn1* expression in brain tissue sections using single-molecule fluorescent in situ hybridization[66] (Fig. 6j and Supplementary Fig. 14e), although the

potential function of the *Nrxn3–Nlgn1* interaction in mediating the interaction between neuron progenitor cells and inhibitory interneurons awaits further exploration. Taken together, by integrating single-cell transcriptome data, our spatial atlas efficiently decodes the spatial proximity and cell–cell communications with STcomm.

## Discussion

Knowledge of organogenesis is crucial for research in regenerative medicine, as the regulatory programs involved in organogenesis are commonly employed to generate cells and tissues both in vivo and in vitro[67]. The spatial architecture of the developing embryo, with its different cell types and anatomical organization is essential for understanding normal development, homeostasis and pathophysiology. While our understanding of embryo organogenesis has been rapidly accelerating, driven particularly by revolutionary single-cell transcriptomics, a key limitation of the single-cell molecular profiling methods is that they operate on disaggregated cells or nuclei so that important spatial information is missing. The emergence of spatial transcriptomic technology has enabled the assessment of cells in their native tissue context, facilitating the identification of location-defined cell types and the understanding of intercellular communications in establishing the body architecture[68,69].

While 2D spatial transcriptomics from the represented embryonic tissue slices has provided important insights into the molecular organization of cell types, the 3D spatially resolved transcriptomic analysis of multiple aligned sections within tissues offers a dynamic angle to dissect spatially defined cell populations, tissue architecture, intercellular interactions along the embryonic axes, including A–P, D–V and L–R. Although not in a highly continuous manner, our spatial atlas, spanning from the head to the tail of mouse embryo organogenesis, covers a wide range of tissue types. Integrated with repeated sampling of additional intercalated sections, the majority of cell types and sub-regions remains unchanged. Therefore, our spatial atlas allows the essential construction of a framework at the whole embryo-level based on 3D spatial coordinates. This represented embryo-level spatial atlas provides an entry to reveal the spatial gene expression profiles that determine the intricate cellular structure of the E13.5 mouse embryo. We also uncovered critical transcriptional regulators for organ and sub-organ development. Importantly, this spatial profiling has enabled the identification of new genes involved in body axis patterning along the anterior–posterior and dorsal–ventral axes, greatly expanding our insights into body patterning during embryogenesis. Leveraging the spatial information, we also delineated the confident L–R interactions between colocalized cell types across the embryo using STcomm.

It should be mentioned that the present study does not encompass a complete 3D reconstruction of an E13.5 embryo, as it would necessitate the examination of a substantially greater number of sections. Moreover, the spatial resolution of this atlas is still not at single-cell level. Therefore, specific cell types or micro-structure of organs

may be hidden at the current resolution. However, our study provided a high-quality resource for dissecting mouse organogenesis development and for developing new bioinformatics pipelines that transition from 2D to 3D analysis[70]. We have built an expandable web portal serving as a spatial transcriptomic resource for further deciphering mouse organogenesis (http://most.ccla.ac.cn). We envision a compiling of more developmental stages with spatial transcriptomics on a large number of consecutive tissue sections or even within the intact tissues to generate a 4D atlas that will greatly deepen our understanding of mammalian embryogenesis and expedite the directed generation of various organs in vitro.

## Methods

### Embryo collection and spatial transcriptome preparation

All animal procedures conducted in this study were approved by the Institutional Animal Care and Use Committee of Guangzhou Institutes of Biomedicine and Health (GIBH), Guangdong. Wild-type embryos at embryonic day 13.5 (E13.5) were collected from C57BL/6JGpt and Gpt:ICR mice aged between 10–12 weeks, that purchased from China GemPharmatech, and embryos images were taken for recording and confirmation of developmental staging.

Collected embryos of C57BL/6JGpt mice were embedded in a tissue-freezing medium (Leica Microsystems, cat. no. 020108926) and stored at −80 °C. The whole embryo tissue was serially cryo-sectioned (Leica CM3050 S) along the craniocaudal axis at 10 μm, and about 1000 sections were harvested. Considering the morphology and uniform sampling, we collected 10 sections from the reference embryo indicated as E1, the position of these 10 sections were 96, 189, 269, 361, 449, 541, 634, 732, 824 and 917 which were named as S1–S10, respectively. For repeat embryo 2 which was indicated as E2, two sections from the head part with similar morphology to S1 and S2 in E1 were selected. For embryo 3 which was indicated as E3, four sections were collected from the body part which was sampled about 50 sections away from S6 to S9 of E1, denoting as female (F) 6.5, F7.5, F8.5, and F9.5. These sections were then used for spatial transcriptomic analysis by modified 10x Genomics *Visium* platform. Briefly, the Visium Spatial Tissue Optimization Kit was used to optimize the permeabilization condition. The ideal embryo tissue permeabilization condition was set to 6 min. Selected sections were stained with 1% cresyl violet solution and imaged using a Zeiss Axio Observer 7 microscope under a 10-lens magnification, then processed for spatial transcriptomics using Visium Spatial Gene Expression Kit (10x Genomics) according to the manufacturer's instructions.

The resulting cDNA was synthesized, amplified and then purified using AMpure X beads. The cDNA library was assessed by Qubit 4.0 Fluorometer and Qsep100 Bio-Fragment Analyzer (Bioptic). The cDNA libraries were sequenced on Illumina Novaseq 6000 system with paired-end 150 bp reads, aiming for 100k raw reads per spot.

**Sex identification of embryos.** For E1 and E2, sex was blindly selected and was determined by the expression of *Xist* and *Ddx3y* from the expression profile after the spatial transcriptome process (Supplementary Fig. 3c, d). To cover both sexes, we collected embryos and determined the sex by examining the expression of *SRY* and *IL3* gene with PCR before the process of spatial transcriptome, and a female embryo was selected as E3.

### Whole-mount in situ hybridization (WISH)

Total RNA was prepared using GEO-seq extraction method[71] from the whole embryos of E13.5 and was further used as a template for preparing probes. Embryos of Gpt:ICR mice were successively fixed, dehydrated and rehydrate in accordance with Yoshihiro Komatsu et al.[72]. The whole-mount embryo was washed with PBS and bleach in 6% hydrogen peroxide for 1 h at room temperature, and incubated with 10 μg/ml proteinase K in PBS for 30 min at room temperature,

then washed by PBS. For pre-hybridization, embryos were placed into the hybridization buffer (5× SSC pH 4.5, 50% formamide, 50 μg/ml yeast RNA, 50 μg/ml Heparin, and 1% SDS) for 1 h at 70 °C. After pre-hybridization, embryos were hybridized with 500 ng/ml RNA probes in a hybridization buffer at 70 °C overnight. Then, embryos were successively washed with pre-hybridization buffer and 1:1 pre-hybridization buffer and 1× TBST buffer (50 ml: 0.4 g NaCl, 0.01 g KCl, 1.25 ml 1 M Tris−HCl pH 7.5, 0.55 g Tween-20) for 30 min at 70 °C. After blocking with 1x TBST containing 0.5% BSA, embryos were incubated with 1× TBST containing 0.5% BSA and 0.1% anti-DIG antibody conjugated to alkaline phosphatase (Sigma-Aldrich 11093274910, 1:2000) at 4 °C overnight, and washed five times with 1× TBST for 1 h at room temperature, followed by washing twice with NTMT (100 mM Tris pH 9.5, 50 mM MgCl2, 100 mM NaCl, 0.1% Tween-20) for 10 min. Finally, embryos were incubated with BCIP/NBT Kit (Cwbio, CW0051S) for staining. The resultant stained slides were imaged with an OLYMPUS SZX16 microscope.

### Single molecular fluorescence in situ hybridization (smFISH)

Fresh embryo was embedded in a tissue freezing medium (Leica) and stored at −80 °C. For validation experiments, RNAscope Multiplex Fluorescent Reagent Kit v2 and PinpoRNATM double-channel Fluorescent Reagent Kit (Pinpo PIF2000) were used on fresh frozen embryo sections 10-μm thick from E13.5 C57BL6J mice in cryostat at −18 °C (Leica CM3050 S), with *Nlgn1* probes(Akoya biosciences 533511) and Opal 570 Reagent Pack (Akoya Biosciences ASOP570), *Nrxn3* probes(Akoya biosciences 505431) *Bcl11a* probe (Pinpo 140251-B1), *Celf2* (Pinpo 140071-B2), *Otp* (Pinpo 184201-B1), *Barhl1* (Pinpo 544221-B2), *Esrrg* (Pinpo 263811-B1), *Nefl* (Pinpo 180391-B2), *Hopx* (Pinpo 743181-B1), *Ttyh1* (Pinpo 577761-B2), *Nsg1* (Pinpo 181961-B1), *Calm2* (Pinpo 123141-B2) and Opal520 Reagent Pack (Akoya biosciences ASOP520), Negative Control Probe (Akoya biosciences 320871, Pinpo P0005), and Positive Control Probe (Akoya biosciences 320881, Pinpo P0002) following the kit instructions. Images were acquired at 20× on an OLYMPUS VS200 microscope.

### Spatial RNA-seq data processing

**Generation of spatial expression matrices.** Quality control and adapter trimming on Raw reads were implemented with fastp-0.21.0[73]. Clean reads were mapped to the mouse reference genome and gene annotations (mm10-3.0.0) using Space Ranger v.1.0.0 (10x Genomics). To obtain only tissue-associated barcodes, spots were manually aligned to the tissue image with the Loupe Browser v.4.0.0 (10x Genomics). Count matrices were extracted by loading the output directory of Space Ranger into Seurat[74,75] (v3.2 and v.4.0.5).

**Data preprocessing.** The expression datasets were filtered with cut-offs at a minimum of 1000 detected genes and a maximum of 10% mitochondrial counts per spot for 10 sections of E1 and E3, except for section 6.5 (F6.5) of E3 which showed relatively lower detected genes (Supplementary Fig. 3), and a minimum of 500 detected genes were applied for F6.5 section. Because of the high quality of our spatial RNA-seq data, only less than 25 spots for each section were filtered. For E1, all spot transcriptomes across 10 sections were merged together. The merged UMI counts were normalized by LogNormalize method with a default scale factor of 10,000 and scaled by the ScaleData function in Seurat with regressing out of sections and number of genes and counts per spot specified by the vars.to.regress argument.

**Variable feature selection.** For variable gene selection, we considered using both high variable genes (HVGs) and spatial variable genes (SVGs) by the following steps. First, we identified 2000 HVGs by vst method from FindVariableFeatures function in Seurat[75]. Second, spatial variable genes were selected by two SVG identification methods. One is implemented through binSpect function in Giotto (R package,

v1.2), a standard general-purpose toolbox for spatial transcriptomic data analysis, which includes a rich set of algorithms for characterizing tissue composition, spatial expression patterns, and cellular neighborhood and interactions[76]. The other is, implemented via SpatiallyVariableFeatures function with Trendsceek method in Seurat[77]. Thus, Spatial variable genes were obtained by intersecting of genes identified by these two methods for each section. We then retrieved both HVGs and SVGs genes as the variable genes for each section. At last, all variable genes in each section were combined together as the final variable gene list after filtering out mitochondrial and hemoglobin genes.

**Dimension reduction, clustering and marker gene identification.** Dimensionality reduction was performed with PCA and then the top 50 PCs were used to create a shared nearest neighbors (SNN) graph and analyzed by Louvain clustering with a resolution of 1.2. UMAP project and visualization in 2D space were also applied to the above SNN graph[14,78]. Differentially expressed marker genes for each cluster were identified by a two-sided Wilcoxon rank sum test using FindMarkers and FindAllMarkers functions.

**Gene Ontology enrichment and spatial domain annotation.** We applied Metascape (http://metascape.org)[79], and clusterProfiler (R package, version 3.14.3)[80] to perform Gene Ontology (GO) enrichment analysis for each group of DEGs and regulon groups. In order to determine the identities of spatial domains and sub-domains, we carefully applied several complementary approaches: (1) we examined the expression of signature genes and enriched GO terms. For most biological systems, there is a scientific consensus on the genes expressed by particular cell types and the annotation based on this works well in a lot of practices; (2) we also double-checked the spot identities based on the deconvolution analysis from single-cell data; (3) we verified the molecular spatial structures with the spatial anatomical structure of the collected image data by referencing annotation of the Emouse Atlas (https://www.emouseatlas.org).

**Data preprocess and spatial domain annotation of female embryo E3.** To explore the batch effect of ST data from E1 and E3, we merged all spot transcriptomes from the 10 sections of E1 and 4 sections of E3 together, and the merged UMI count matrix was normalized and scaled following the same process in the Data preprocessing part. Since minimum batch effects were identified, we annotated the major spatial domain of 4 section dataset of E3 by label transferring using the modules of FindTransferAnchors and TranferData in Seurat referenced by the ST dataset of E1 (Supplementary Fig. 3e–g). For the subcluster annotation of the D5-Visceral organ with smooth muscle, after extracting the spots from this major spatial domain of both datasets, the label transfer process was again performed to annotate the subdomain of visceral organs in the E3 dataset.

**Integrative analysis of MOSTA and reference data.** E13.5 S1 data of MOSTA was obtained from https://db.cngb.org/stomics/mosta. The data integration of the MOSTA dataset and the reference E1 dataset was performed using FindIntegrationAnchors and IntegrateData. Label transfer was also applied to annotate the MOSTA dataset with FindTransferAnchors and TranferData modules in Seurat.

**Sub-regional identification and annotation for heart domain.** We extracted spots from the D19-heart domain, and processed the UMI counts following the process in the Data preprocessing part. The variable genes were defined as the combination of 500 HVGs with 'vst' method and the top 100 marker genes of 6 cell types of the heart during development including cardiomyocytes of atrial and ventricular, epicardial, fibroblasts and endothelial cells of vascular and endocardial retrieved from Feng et al.[43]. Dimensionality reduction with

PCA and clustering by Louvain with a resolution of 1.1 were performed. All downstream steps followed the Spatial RNA-seq data processing pipeline.

**Spatial expression pattern analysis of congenital heart disease (CHD) genes.** To examine the spatial expression pattern of CHD-genes, we first retrieved the curated known CHD genes from Feng et al.[43] and then detected and clustered the spatially correlated genes using detectSpatialCorGenes and clusterSpatialCorGenes function in Giotto v1.2. The average spatial expression of clusters was calculated and visualized. We searched genes from literatures that are associated with CHDs or the CHD-associated risk factor knowledgebase (http://www.sysbio.org.cn/CHDRFKB/) for certain spatial patterns[43].

**Identification of common or sex-specific Gonad marker genes.** To detect the sex-specific or common genes during gonad development after sex determination, we collected sections from both female and male embryos covering the region of visceral organs. We calculated the gonad marker genes by DEG analysis with the Wilcoxon rank sum test using FindMarkers function for male and female datasets, respectively. We also identified the sex differential genes in Gonad by performing DEG analysis in the spot transcriptomes of gonads from both sexes. We thus defined the common gonad marker genes as the intersection of gonad marker genes in males and females.

## TF and Regulon analysis

To infer transcription factors and the gene regulatory network (regulon) in our spatial transcriptome data, we applied SCENIC (Single-Cell rEgulator Network Inference and Clustering) pipeline with pySCENIC (v0.10.3), a lightning-fast python implementation[18].

The procedure contains three main steps: (1) co-expression modules between TFs and the candidate target genes were first identified base on the correlation of normalized gene expression across all sample spots by GRNBoost2 with default parameter settings. Genes expressed in less than 10 spots were filtered. (2) co-expression modules were then further pruned by keeping only direct targets of TFs based on motif discovery by RcisTarget. Thus, modules composed of TF and TF direct target genes were defined as a regulon. (3) The Regulon Activity Score (RAS) for each spot was calculated through the area under the recovery curve by AUCell and the regulon activity for each spatial domain was computed as the mean activity of the corresponding spots.

## Calculation of regulon-specific score (RSS) in spatial domains

We calculate the spatial domain specificity score of a regulon as described in Suo et al.[19]. Briefly, the RAS in all spots was normalized as a probability distribution and the indicator vector of a spot belongs to a specific spatial domain or not was also normalized as a probability distribution. Next, the Jensen–Shannon divergence (JSD) was computed between these two probability distributions. Finally, the RSS was calculated by converting the JSD to a similarity score. The selected regulons with top RSS for each spatial domain indicated their specificity and essentialness in the corresponding spatial domain.

## Regulon module analysis

To examine the co-regulation of TFs, we performed regulon module analysis as described previously[19] which involves two steps. First, we calculate the connection specificity index (CSI) for each pair of regulons following the instruction in[29], which is a context-dependent metric used for identifying specific associating partners. Next, regulon modules were identified by clustering the regulons with hierarchical clustering based on the Euclidean distance of the CSI matrix.

To study the relationship among different regulons, we also build the regulon co-activation network based on the CSI matrix with a threshold of 0.85 to filter weakly connected regulons and visualized the network by Cytoscape (v3.9.1)[81]. The mean regulon activity scores

of each module were calculated by averaging the RAS of all regulon members that belong to this module across all spots. The top correlated spatial domain for each module was identified based on the spatial distribution of mean regulon activity scores.

## Spatial pattern of spinal cord analysis

To systematically explore the spatial patterning from the anterior–posterior (A–P) axis, we conducted the pseudo-space analysis in the spots from the spatial domain of the hindbrain and spinal cord. All Hox genes expressed in the dataset were used for reconstructing the pseudo-axis by Monocle 2 (v2.18.0)[82]. The expression of typical A–P patterned Hox genes was smoothed in extracted spots which were first ordered along the sections from anterior to posterior and then ordered by predicted pseudo-space within each section.

To identify A–P-related genes besides Hox genes, we applied the following steps: (1) identified significant differential genes among the A–P slices by differentialGeneTest function in Monocle 2 with the pseudo-time formula. (2) to select specific A–P patterning genes, we then calculated the Pearson correlation between the expression of the above-identified differential genes and different consecutive combinatorial patterns of A–P sections. It included a total of 35 consecutive combinations from S2 to S10. The union of genes with the top 20 PCC for each combinatorial pattern was selected as strong AP-related genes.

To examine the spatial patterning along dorsal–ventral (D–V) axis, we retrieved a list of marker genes that were used to identify D–V domains of neuronal progenitors and neuron clusters from Delile et al.[45]. As our 10× Visium spatial transcriptomics are not in single-cell resolution, we separated the predefined dorsal ventral marker genes into Dorsal (D), media (M) and ventral (V) domain-related gene sets for both the inner progenitors and the outer neuron regions. In brief, for neuronal regions, D marker gene sets were defined by combinatorial markers of dI1–dI6, M marked by V0–V2b and V marked by Mn and V3. Similarly, for neuron progenitor regions, D marker gene sets include combinatorial markers of RP and dp1–dp6, M includes markers of p0–p2, V includes pMN, p3 and FP maker genes. Then, the activity score of each region-related gene set was calculated by AUCell (v1.8.0)[18]. For visualization, the activity scores were scaled and scores lower than the binary assignment score were set to 0 in the outer neuron region. In the progenitor region, the z-score lower than 2.5 were set to 0.

To identify D–V-related genes, we divided the spinal cord region into dorsal, middle and ventral domains according to the D, M, V region-related activity score. Differential expression analysis was conducted among these three regions by FindallMarkers Function in Seurat. Significant DEGs with adjusted $p$-value < 0.05 were considered as D–V-related genes. For neuronal progenitor genes identification, we calculated differential expressed genes by comparing D17-ependyma and D16- medulla oblongata and spinal cord on sections from S4 to S10 based on the union of the marker genes of D16 and D17.

## Regulon analysis for spinal cord patterning

To identify the regulons with potential A–P patterning, we calculated the correlation of RAS and the predicted pseudo-space ordering in the spots of the spinal cord. The A–P-related regulons were defined as with absolute PCC higher than 1.5 × SD PCCs and maximum RAS greater than 0.2. For D–V-related regulons, we performed differential analysis on RAS of D, M, V parts by Wilcoxon test implemented by FindMarkers Function in Seurat and retained regulons with maximum RAS > 0.2.

## Signaling pathway activity analysis in spatial transcriptomic data

Signaling activity scores were computed using the *AddModuleScore* function from Seurat based on the literature-curated critical development-related signaling signatures, including BMP, Wnt, Nodal, Fgf, Hedgehog, Hippo and Notch signaling pathway genes. Briefly, the score of each spot was computed as the average expression of each signaling signature subtracted by the aggregated expression of random control gene sets[83].

Cell cycle phase activity scores were calculated for all spots by CellCycleScoring function from Seurat function with mouse cell cycle-related genes retrieved from Giladi et al.[84].

## Spatial deconvolution of single-cell types

To spatially map cell types of mouse organogenesis, we used the single-cell dataset of Trajectories Of Mammalian Embryogenesis (TOME, downloaded from http://tome.gs.washington.edu/) at the E13.5 stage as reference[1], and performed deconvolution for each section by the robust cell-type decomposition (RCTD, v1.0.4), which is a supervised learning method to accurately decompose the spatial transcriptomic mixtures for each pixels by using a scRNA-seq reference containing cell-type classifications[58]. Before running the RCTD, we excluded Hemoglobin and Mitochondrial genes from both single-cell and our spatial transcriptome datasets. The method of multi-mode was selected to perform deconvolution analysis on our spatial data with default parameters, except for CELL_MIN_INSTANCE = 25, UMI_max = 2e + 08. We extracted the cell-type deconvolution weights of each spot for downstream analysis and visualization.

For cell type deconvolution of the heart domain, we downloaded the scRNA-seq dataset of the heart from GSE193346 and extracted the data at the E13.5 stage, and performed deconvolution as the above process.

## Cell type co-localization and cell–cell communication analysis

To study cell-type interactions in the spatial microenvironment, we develop an analysis workflow named STcomm (Fig. 5c). It combined the spatial cellular colocalization of single cells and ligand-receptor co-expression from spatial transcriptomic data and cell–cell communication based on the expression of ligand–receptor pairs in the single-cell transcriptome. STcomm has the following steps:

(1) we performed deconvolution analysis with RCTD to generate the distribution of different cell types in each spot. Spatial-wide cell-type colocalization network was created base on the PCC of cell-type weights. The co-localization cell type pairs were obtained with PCC > 0.06 and adjusted $p$-value < 0.05[85]. These satisfied cell-type connections were visualized by Cytoscape[81].

(2) we extracted the expression of each pair of LRs in the ST dataset and calculated the spatial co-expression level of LR pairs as $R_{LR} = L(\exp)*R(\exp)$. The ligand–receptor information was extracted from the CellChatDB.mouse database[86].

(3) for each spatially co-localized cell type pair, we binarized it based on the confident Boolean value of cell-type weights generated in RCTD and only reserved co-localized cell type pairs identified in step (1). Meanwhile, we also binarized the co-expressed LR pairs accordingly to evaluate whether $R_{LR} > 0$. Finally, we built the 2 × 2 contingency table based on binarized cell type co-localization and LR co-expression information and performed Fisher's exact test followed by Benjamini–Hochberg correction to statistically identify spatially enriched co-expressed LR pairs in the co-localized cell type pairs with adjusted $p$-value < 0.05.

(4) we ran cell–cell communication by CellChat (v1.4.0) with the default parameters on the scRNA-seq dataset (of E13.5 TOME) to get the probability and significance of spatially co-localized cell-type communications on all L–R pairs. Only these L–R communication results which were identified as spatially enriched co-expressed L–R pairs within co-localized cell type pairs were finally retained.

 

## Web service and 3D illustration

The Mouse Organogenesis Spatial Transcriptomic dataset can be interactively explored at our website (http://most.ccla.ac.cn/), which was constructed to navigate the spatial atlas of all 10 sections from E13.5 embryo using Shiny (v1.7.4) in R. This web service provided four parts for data exploring, to explore the spatial domains of all sections based on molecular signatures in the 2D and 3D format and 3D embryonic model by the 'Spatial Domain Explorer', to search for the spatially resolved gene expression by the 'Spatial Transcriptomics Explorer' in 2D and 3D space, to identify the spatial regulon modules on each section by the 'Regulation' in 2D and 3D space, to retrieve the spatial pattern of co-expressed genes by the 'Gene Pattern Explorer'. To illustrate the molecular characteristic and spatial domains of E13.5 embryo in a three-dimensional whole embryo scale, we retrieved the images of TS22 corresponding to the stage of E13.5 from eMouse atlas (www.emouseatlas.org), and then registered our 10-section images manually with the matched images in TS22 using Adobe Photoshop, and stacked all these images together by ImageJ (v1.53c).

## Statistics and reproducibility

Appropriate statistical tests are performed for each analysis and specified in the respective figure legend. All WISH/RNAScope experiments were repeated on at least three independent biological replicates unless indicated otherwise and representative images were presented in the manuscript. No statistical method was used to predetermine the sample size. The experiments were not randomized or blinded. The investigators were not blinded to allocation during experiments and outcome assessment.

## Reporting summary

Further information on research design is available in the Nature Portfolio Reporting Summary linked to this article.

## Data availability

The mouse organogenesis spatial transcriptomic data and corresponding section images generated in this study have been deposited in the GEO database under accession code GSE237308 and the NODE (The National Omics Data Encyclopedia) database under accession code OEP003721 and can be explored at the web portal (http://most.ccla.ac.cn). The MOSTA data at stage E13.5 used in this study are available in the CNGB database under accession code CNP0001543 and the processed data are available at https://db.cngb.org/stomics/mosta. The single cell dataset of heart for deconvolution analysis used in this study is available in the GEO under accession code GSE193346, and TOME data used in this study are available under accession code GSE186068. Source data are provided with this paper.

## Code availability

Source code for STcomm is available on github at https://github.com/gpenglab/STcomm and Zendo at https://doi.org/10.5281/zenodo.7988217.

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

## Acknowledgements

This work was supported in part by the National Key R&D Program of China 2018YFA0801402 to G.P., the "Strategic Priority Research Program" of the Chinese Academy of Sciences XDA16020404 to G.P., National Natural Science Foundation of China (32270854, 32161160322 to G.P. and 32100483 to G.C.), Guangdong Basic and Applied Basic Research Foundation (2019B151502054 to G.P., 2019A1515110985 to G.C., 2023A1515011783 to S.S. and 2020A1515110517 to F.Q.), Science and Technology Program of Guangzhou 202102080293 to F.Q., Frontier Research Program of Bioland Laboratory (Guangzhou Regenerative Medicine and Health Guangdong Laboratory, 2018GZR110105013), Jiazi Research Innovative Project of Bioland Laboratory (2019GZR110108001), Science and Technology Planning Project of Guangdong Province 2020B1212060052 to G.P. We thank X.L. Peng, X.Z. Zhai and L.F. Liu for experimental support. We thank P. Tam, N. Sheng, G. Bai, and N. Jing for the discussion and critical reading of this study.

## Author contributions

G.P. and F.Q. designed the study. G.P., S.S., G.C. and F.Q. supervised the project. F.Q., W.L., X.R., F.L., and R.Z. analyzed the data with contributions from J.K. and Z.Z. J.X., G.C. and X.M. performed the experiments with contributions from L.Q., J.Z., X.L. X.Z. and M.W., G.W., J.C. and D.P. provide important reagents and suggestions. G.P. and F.Q. wrote the paper with the help of W.L., G.C., S.S. and J.X.

## Competing interests

The authors declare no competing interests.
