## [Peer Review File · Nature Communications]

Three-dimensional molecular architecture of mouse organogenesisREVIEWER COMMENTS

Reviewer #1 (Remarks to the Author):

In this manuscript, Qu et al. reported a holistic spatial transcriptome atlas of the mouse embryo at embryonic day 13.5 in a 3D dimension using spatial transcriptomic analysis. Based on 10 continuous slices of mouse embryo samples, they uncovered critical and potential novel transcriptional regulators for the development of tissue-specific functions, and also enabled the characterization of transcriptome heterogeneities in the body axis along the anterior and posterior or dorsal and ventral view. To investigate the cell type heterogeneity in one spot, they used scRNA-seq data to perform cell type deconvolution analyses. Additionally, they developed a spatial cell-cell communication method named STcomm which can identify the spatially co-localized cells and the significantly enriched ligand-receptor (L-R) pairs between those cell types.

Overall, the study provides a comprehensive spatial expression landscape of the mouse embryo from the 3D view with high resolution. Although without consecutive sampling time points to compare the transition of cell types between developmental stages, the data presented here are still of great value to the developmental biology field. The reviewer has several comments to improve the manuscript, mainly about the methodology details.

Major Comments:

1. In the A-P slices analyses, the authors identified distinct A-P patterning genes through differential gene analysis according to the predicted pseudo-space ordering and applied gene correlation analysis between spots and AP slices to select top related A-P patterning genes. I am wondering how the authors defined the correlation cut-offs and how many genes the authors choose as A-P-related genes? Are there any TFs or TF regulons that have different enrichment patterns along the A-P axis? Similarly, are there any TFs or TF regulons that show differences along the D-V axis? Finally, the authors should explain why the analysis is only limited to the spinal cord regions.

2. The authors claimed that the current datasets present a 3D view of developing mouse embryos. However, the authors did not provide any details about how different slides were registered to reconstruct the 3D view. Also, will the registration strategy influence the identified A-P-specific genes?

3. To investigate spatial cell-cell communication, the authors developed a new method named STcomm using deconvolution co-localized cell type information and scRNA-seq L-R (ligand-receptor) expression information. The authors should provide more details about the methods. For example, the authors mentioned that “we calculated co-localized cell types pairs and performed binarization according to the decomposed cell-type weights”. What is the cell type weight cut-off used to define co-localized cell type pairs within spots? Will it change the result significantly if different cell type cut-offs are used? How are the identified LRs ranked in different spots? By the p-value of CellChat?

4. The Mouse Organogenesis Spatial Transcriptomic dataset website (<http://most.ccla.ac.cn/>) the authors provided at the end of the Methods section is invalid (Internal Server Error).

Minor Comments:

1. In Extended Data Fig. 7b, the posterior floor plate cell type is located in only one to two spots for sections across the body trunk after spatial mapping, while the identified marker genes in TOME dataset such as *Slit1*, *Ntn1* and *Shh* located in more spots on the edge of the slides, the inconsistency should be explained.

2. In Extended Data Fig. 3d, the authors show D14, D3 and D4 domains top regulons AUC activity score, while the top regulons TF expression on the right seems didn't match well. This should be explained or using better examples.

3. In the second paragraph of the section 'Construction of a spatial transcriptome atlas of embryo organogenesis at E13.5', the authors wrote, 'For example, D6-hepatic parenchyma, specifically expressing Afb and Apoa2'. Afb should be Afp.

4. In Figure 3a, the cell color inside Cluster 8 is inconsistent resulting in the different cell color in the left two most sections in figure 3b.

Reviewer #2 (Remarks to the Author):

In this study, Qu and colleagues performed spatial transcriptomic assays on 10 sections of mouse embryos staged at E13.5, to generate a 3D atlas. Leveraging this dataset, they next identified 19 distinct spatial domains globally and their key TFs, and then focused on several interesting questions regarding development, including subdomains of the visceral organs, and A-P/D-V axis of spinal cord. Moreover, they integrated the dataset with other single-cell RNA-seq datasets to further deconvolute the cell-type heterogeneity of each spatial spot.

Although we find that study interesting, we do have a number of major concerns:

Major comments

1) There are several concerns regarding the "resolution" of the current spatial transcriptomic dataset:

- Most of the spatial domains (D1-D19) were defined in an "organ" or "tissue" scale, e.g. heart. Then, authors nominated key TFs or genes for individual spatial domains. However, the resolution might not be sufficient. People could perform RNA-seq on dissected tissue samples, or single-cell RNA-seq, to obtain the similar transcriptional information. We don't see a significant improvement by performing spatial transcriptomics documented.
- Most of the spatial domains or subdomains were purely identified by transcriptional heterogeneity. Do the author have anatomical features to verify those annotations? e.g. some type of imaging data.
- The authors indicated "3D atlas" many times in the manuscript, however, I didn't see any analysis that was across different sections. It's not clear to me why the authors termed it 3D as opposed to just a limited set of sections. 3D implies a model that is really contiguous in all three dimensions.

2) The authors identified the key TFs and TFs modules for individual spatial domains. Some potential concerns:

- It would be better to benchmark those TFs or TF regulons in some ways. At least authors should show which TF is known, which TF are relatively new, and which are unexpected.
- The TF modules are very interesting, but it necessarily needs to dig deeper. Are those TF-TF interactions reliable or not? What are the potential molecular mechanisms (one or two examples)? Otherwise I don't think it's very informative compared to the left sections of the manuscript. Currently, without any experimental validation or literature supporting, all the conclusions are more likely speculated from kinds of "co-expression" analysis.
- I feel the second half of the manuscript, including the subdomains of the visceral organ, A-P/D-V axis of spinal cord, and integrating with other datasets are way more interesting. I would suggest slightly reorganizing the manuscript to emphasize the second half.

3) A suggestion, in Fig.1d, is it possible to make a 3D visualization of all the 10 sections

even though it's not in a "continuous" way? For example, domain 14 is distributed in three different sections, it might be hard for the audience to build a "3D" view of how domain 14 is distributed in a real whole embryo. I imagine something like the one in the subpanel (1) of the panel (a) would be much more straightforward.

4) Most of the identifications are way "safe". It doesn't sufficiently show how useful such an interesting dataset is.

5) Could the author try to compare the current data to some other spatial transcriptome data during mouse organogenesis, e.g. MOSTA or sci-Space?

Minor comments

1) In Fig. 2a, only some of the TFs are highlighted on the right of the heatmap. How were they selected?

2) In Fig. 2a-b, I am curious, does higher RAS of TF regulon also guarantee higher TF expression? It would be interesting to plot gene expression for those TFs as well, i.e., making the same heatmap but using TF expression instead of RAS.

3) It's not clear to me - "By computing the averaged RAS score for each TF regulon module ... in particular tissue regions" (Page6, 182-184). How were the region-specific modules of TF regulons identified? More specifically, in the Fig.2c, how was each module assigned with one or more spatial domains?

4) In Fig. 4a-b, the colormap is not quite distinct for different sections, S2-S10. I suggest using a different colormap. Also, Hox genes are very specific to the anterior and posterior, but they don't show a clear pattern in the central region, could the authors comment on it?

Reviewer #3 (Remarks to the Author):

Mammalian embryogenesis is a well-orchestrated process involving a series of cell proliferation and differentiation processes. Understanding the cellular dynamics and spatial distribution of developing cells during organogenesis is critical in developmental biology. In this study, Qu et al used 10X Visium spatial transcriptomics technology to analyze cross sections of an E13.5 stage mouse embryo. The authors carried out extensive exploration of the spatial patterning of gene expression during embryogenesis, including tissue type characterization, TF network and regulation inference, A-P axis expression dynamics, and cell type deconvolution. Meanwhile, the authors proposed to use Visium spot as the natural aggregation of different cell types to test for cell-cell communication through ligand-receptor pair analyses. Overall, the computational analysis is comprehensive and well-executed, though the study itself still has some major weaknesses, detailed below.

First, the whole work was based simply on 10X Visium experiments using 10 tissue sections, out of ~1000 sections from a single mouse embryo at E13.5. The highly selected tissue sections indicate strong sampling bias in projecting spatial gene expression patterns for the embryo. It is almost impossible to reconstruct a 3D transcriptomic model of the E13.5 embryo. For this reason, it is very hard to call this dataset a '3D' spatial transcriptomic atlas/resource, nor being truly useful for the community. The authors did validate the data quality with a second embryo with two sections. However, this validation data does not resolve the intrinsic bias in the main data, let alone the poor quality in the second embryo. To resolve this sampling bias, the easiest solution is to unbiasedly increase the number of

embryo sections to the level that could enable the reconstruction of a 3D transcriptomic model.

Second, despite very comprehensive analyses, the bulk of the results are descriptive and just validate what is already known in the specific areas. Regardless, the data could have led to non-descriptive new discoveries. The study does not generate new biological discoveries that could potentially demonstrate the power of ultimately building a comprehensive and single-cell resolution 3D spatial transcriptomic atlas. The authors claimed to discover spatial patterns of new genes – which are supposed to be expected – these results should be backed up with more solid validation with e.g. whole embryo ISH assays and/or lacZ reporter assays, and additionally examine their biological function in case studies. These validations are more necessary considering that Visium ST technology only analyzes mini-bulk transcriptomes, and bears strong technical noise including leaking transcripts across the section.

Third, what is the sex of this embryo? This is critical information for mouse embryos developed beyond E11 when the gonad starts to have sex-specific differentiation. The authors claimed in Line 262-264 that *Lefty2*, *Tex19.1*, and *Dppa3* are gonad-specific genes (Fig. 3c). Are they male- or female-specific gonad genes? It will be interesting to see the expression pattern of sex-specific gonad gene expression, such as *Sox9*. For this reason, the authors should analyze both male and female embryos for a comprehensive resource, and again, could also use developing gonads as a system to demonstrate the power of ST technologies for studying developmental biology questions.

Other concerns:

The mouse embryo sections have been well-characterized. However the authors did not seem to use their H&E staining images for cross-reference analyses. Since the embryo is not straightly A-P aligned, the H&E staining helps bridge the actual tissue and the ST data. To the least, presenting the corresponding H&E staining images for each analyzed section helps to indicate tissue quality in the ST experiment. Fig. 1a schematic does not seem to reflect the actual sections used in this study.

Line 131, in the text, D17-ependyma was presented as ubiquitous existence across the assayed embryo sections. However, in Extended Fig. 2b, it was only evident in the head sections. The authors need to examine carefully if any misinterpretations of the results. Fig. 2c, how was the value of molecular mean activity calculated? Meanwhile, D4, D5, and some other groups seemed missing in this analysis. A similar problem happened in Fig. 4a-c where sections 1&9 or 1&10 were missing.

Extended Fig. 1i and n, there seem to be typos in the tick labeling to distinguish E1 and E2.

Response to reviewers

Summary of major new data

We would like to thank the reviewers for their valuable inputs. We provided a revised version of the manuscript with a substantial body of new experimental data and analysis that further substantiate our major conclusions.

1. We collected 4 more sections of spatial transcriptome including the developing gonad from a female embryo at the same E13.5 stage by the same 10X Visium platform. These data showed a good match with our 10-section spatial dataset and indicated that the collected 10 sections have a reasonable representation of the whole embryonic tissues not only at the anteroposterior but also at distal-proximal axis, reassuring that our dataset would serve as a rich resource for investigating the 3D molecular structure of mouse organogenesis.
2. We performed more experimental validation of identified marker genes by FISH RNAscope and Whole mount ISH, together with thorough literature reviewing, to highlight the specificity, robustness and utility of our spatial data and comprehensive analysis.
3. We enhanced the 3D visualization by reconstructing a 3D embryonic model and illustrated the spatial view of identified spatial domains within the virtual embryo model. We also improved the 3D illustration of spatial gene expression and regulon activities in our web portal. All these efforts highlighted the 3D recapitulation of molecular activities through meticulous alignment and collective integration of multiple spatial-transcriptome (ST) sections.
4. As suggested by the reviewer and the editor, we performed analysis on the comparison of sex specification on the developmental gonad and identified common and sex specific gonad marker genes, which would shed light for sex differentiation studies.
5. We also demonstrated the expandability and high spatial resolution of our ST dataset by dissecting the spatial sub-organ regions of embryonic heart. Besides for identifying the typical heart anatomical regions and mapping the spatial distribution of cell types with single-cell data of heart development, we also correlated developmental signatures to the congenital disease.

Reviewer #1 (Remarks to the Author):

In this manuscript, Qu et al. reported a holistic spatial transcriptome atlas of the mouse embryo at embryonic day 13.5 in a 3D dimension using spatial transcriptomic analysis. Based on 10 continuous slices of mouse embryo samples, they uncovered critical and potential novel transcriptional regulators for the development of tissue-specific functions, and also enabled the characterization of transcriptome heterogeneities in the body axis along the anterior and posterior or dorsal and ventral view. To investigate the cell type heterogeneity in one spot, they used scRNA-seq data to perform cell type deconvolution analyses. Additionally, they developed a spatial cell-cell communication method named STcomm which can identify the spatially co-localized cells and the significantly enriched ligand-receptor (L-R) pairs between those cell types.

Overall, the study provides a comprehensive spatial expression landscape of the mouse embryo from the 3D view with high resolution. Although without consecutive sampling time points to compare the transition of cell types between developmental stages, the data presented here are still of great value to the developmental biology field. The reviewer has several comments to improve the manuscript, mainly about the methodology details.

Our response: We thank the reviewer for the encouraging comments and acknowledging the value of this study.

Major Comments:

1. In the A-P slices analyses, the authors identified distinct A-P patterning genes through differential gene analysis according to the predicted pseudo-space ordering and applied gene correlation analysis between spots and AP slices to select top related A-P patterning genes. I am wondering how the authors defined the correlation cut-offs and how many genes the authors choose as A-P-related genes?

Our response: Thanks for pointing out these important questions. We previously have briefly described the analytical pipeline in the methods part and we have updated with more details. To identify specific spinal cord A-P patterning genes, we first applied differential gene analysis according to the predicted pseudo-space ordering, and then calculated the correlation between gene expression and different consecutive combination of spinal cord spots along A-P sections. It included a total of 35 consecutive combinations from S2 to S10 (e.g. two consecutive sections include S2-S3, S3-S4..., three consecutive sections include S2-S3-S4..., and so on). To obtain the final set of A-P pattern genes, we selected genes with top 20 PCC under each combination. We obtained 77 genes through union of these gene sets which was shown in the heatmap plot of Extended Data Fig.8a. 21 Hox genes which are well-known Hox A-P pattern genes were identified in this list, highlighting the fidelity of this gene list and the analysis strategy.

Are there any TFs of TF regulons that have different enrichment patterns along the A-P axis. Similarly, are there any TFs or TF regulons that show differences along the D-V axis?

Our response: These are good questions. We now added the TF regulon analysis along the A-P and D-V axes. Indeed, we identified a set of TF regulons which show the A-P pattern, and similarly Hox regulons are the mostly identified ones (Revised Extended Data Fig. 9a-b). Among these, 3 Hox Family TF regulons showed strong activity in posterior regions, and 7 TF regulons have relatively high activity in the central to posterior regions including Pitx3(+) and Sox5(+) which involved in neuronal development [doi: 10.1007/s00441-004-0943-1, doi: 10.3389/fnmol.2021.654031]. The rest of TF regulons were strongly activated in the anterior regions such as Evx1(+), which plays a critical role in neuronal specification [doi: 10.1038/srep26657].

For D-V patterned regulons, we applied differential activity analysis on Dorsal, Media and Ventral spinal cords, and excluded regulons with the maximum of activity score less than 0.2 in spinal cord spots. We revealed a total of 28 TF regulons with differential activity along the D-V axis (Revised Extended Data Fig.9c-d). As visualized by spatial plots, these regulons showed a clear D-V patterns, such as Mnx1(+) and Npdc1(+) with high activity in dorsal and Lbx1(+) in ventral. Hoxc9(+) was identified with high activity in the middle region. We have included these results in the revised manuscript.

Finally, the authors should explain why the analysis is only limited to the spinal cord regions.

Our response: Thanks for this suggestion. As a classical question of developmental biology, understanding the spatial patterning of craniocaudal (A-P), dorsoventral (D-V) and radial axes Center-Outer are crucial to study the process of embryonic development. Our spatial atlas retained the real spatial location and covered all three axes of spinal cord; thus, we conducted a detailed analysis of spatial patterning in spinal cord, and found known and new axes related genes and TF regulons as mentioned above. Recently, several single-cell studies also tried to speculate the spatial patterning of spinal cord development [e.g., doi: 10.1242/dev.173807; doi: 10.1038/s41593-020-00795-0; 10.1038/s41556-023-01108-w]. However, these studies either lack of spatial location information or lack of whole transcriptome-wide analysis, which showed the advantages of our spatial atlas.

For A-P patterning (craniocaudal), Hox gene code is well studied in hindbrain and spinal cord to establish the regional identities, and we collected hindbrain and spinal cord tissues from section 2 to 9. Thus, we explored new A-P-related genes in these tissues. Furthermore, we also examined the novel genes in dorsal-ventral (D-V) patterning and radial axis (medial-lateral) patterning which are related to distinct neuronal subtypes differentiation. Notably, Celf2 and Bcl11a were highly expressed in dorsal side, whereas Esrrg and Nefl were enriched in ventral side of spinal cord. Otp and Barhl1 were up-regulated in the medial region along D-V axis. Ttyh1 and Hopx were expressed in the inner regions, while Nsg1 and Calm2 were highly expressed in the outer layers which may be related to differentiated neuronal cells from neural progenitors. Those new genes were confirmed by FISH RNAscope (Figure R1, Figure 4 and Extended Data Fig.8c)

Figure R1. Dorsoventral and radial patterning related genes in spinal cord
A-F, The expression pattern of dorsoventral axis related genes in spinal cord. G-H, The expression pattern of medial-lateral axis related genes in spinal cord. Some marker genes identified by DEGs analysis in our dataset and were validated by FISH RNAscope.

2. The authors claimed that the current datasets present a 3D view of developing mouse embryos. However, the authors did not provide any details about how different slides were registered to reconstruct the 3D view. Also, will the registration strategy influence the identified A-P-specific genes?

Our response: We're sorry that our website was not accessible during the review period and therefore we were not able to show the 3D embryo model with the inserted 10 slides. We now fixed it and also added the 3D embryo model to show the distribution for each spatial domain

(Supplementary Video Files, www.most.ccla.ac.cn). For registering different slides, as mentioned in the method, we reconstructed a standard E13.5 3D embryo model by stacking the images of Embryo TS22 corresponding to the stage of E13.5 retrieved from eMouse atlas (<https://www.emouseatlas.org>). Then we retrieved 10 standard slides from TS22 3D embryo model which matched to our E13.5 embryo sections according to the recorded position and the spatial anatomical structure of given slides. We tried several automated registration methods including Matlab control point selection (MathWorks) and ImageJ plugin elastic registration, while these methods were not able to provide a good match of our slide with the standard slides, thus we manually registered the paired slides by Adobe Photoshop and replaced the corresponding matched slides from 3D embryo model with our 10 sequenced slides and stacked all these slide images together and created a 3D view with ImageJ.

We also tried PASTE [doi: 10.1038/s41592-022-01459-6], a method to model transcriptional similarity and physical distance between spots for integrating multiple slices but did not achieve a good conformity. We reasoned that there could be sophisticated variances between slices of big distances and changed structure.

As the registration strategy was only used for illustrating our slides in the 3D model and did not include molecular alignment. This process does not influence the identified A-P specific genes.

Figure R2. The image registration to standard slides from TS22 embryo of eMouse Atlas for S1, S2, S6 and S10.

3. To investigate spatial cell-cell communication, the authors developed a new method named STcomm using deconvolution co-localized cell type information and scRNA-seq L-R (ligand-receptor) expression information. The authors should provide more details about the methods.

For example, the authors mentioned that “we calculated co-localized cell types pairs and performed binarization according to the decomposed cell-type weights”. What is the cell type weight cut-off used to define co-localized cell type pairs within spots? Will it change the result significantly if different cell type cut-offs are used?

Our response: We apologize for the unclear description of STcomm method in the original submission. We have revised the schematic workflow in Figure 5c and added detailed descriptions in the method part.

For the cut-offs of defining co-localized cell type pairs in a given spot, we applied the confidence value generated in Robust Cell Type Decomposition (RCTD) [doi: 10.1038/s41587-021-00830-w] which converted the cell type weight into a binary value. We then applied Fisher’s exact test on the binarized co-localized cell type value and the binarized spatially co-expressed LR pairs to retrieve the enriched LRs pairs in corresponding cell type pairs.

At first, since the confidence value generated by RCTD was evaluated by simulation results, we think it is relatively reliable. Secondly, We also benchmarked several different cut-offs to determine the confidence value in RCTD, and examined the difference of retained LR pairs in co-localized cell type pairs under these cut-offs. We found though the Boolean value of co-localized cell type slightly changed, the retained LR pairs do not change significantly. We thus reasoned the method using this approach for determining the co-localized cell type pairs is stable.

How are the identified LRs ranked in different spots? By the p-value of CellChat?

Our response: We first used the above described method to obtain spatially co-expressed LRs in co-localized cell type pairs based on ST data (revised Fig 5c). To rank these spatially enriched LRs, we first applied the p-value of CellChat to filter the non-significant LRs based on the corresponding scRNA-seq data, and then ranked the LRs by the communication probability calculated by CellChat. We finally focused on LR pairs with the high score for further exploration.

4. The Mouse Organogenesis Spatial Transcriptomic dataset website (<http://most.ccla.ac.cn/>) the authors provided at the end of the Methods section is invalid (Internal Server Error).

Our response: We are sorry for the error of the website server. Now we have fixed it. To help users to explore our spatial atlas comprehensively, our web portal includes four major functional modules. The “Spatial Domain Explorer” serves for exploring the spatial domain of all sections in the 2D, 3D formats and in 3D embryonic model. The “Spatial Transcriptomics Explorer” serves for exploring the spatial expression pattern of genes in mammalian embryo organogenesis in 2D and 3D space. The “Regulation Explorer” serves for exploring the spatial activity score of regulons in mammalian embryo organogenesis in 2D and 3D space. Finally, the “Gene Pattern Explorer” serves for retrieving genes that share similar expression patterns of a query gene in space by Pearson correlation analysis. Meanwhile, we also packed the 3D embryo models which include each spatial domain in the Supplementary Video Files.

Minor Comments:

1. In Extended Data Fig. 7b, the posterior floor plate cell type is located in only one to two spots for sections across the body trunk after spatial mapping, while the identified marker genes in TOME dataset such as *Slit1*, *Ntn1* and *Shh* located in more spots on the edge of the slides, the inconsistency should be explained.

Our response: Thanks for the suggestion. We looked into this problem. First, we performed cell type deconvolution by RCTD [doi: 10.1038/s41587-021-00830-w] with E13.5 mouse single cells derived from TOME dataset as the reference [DOI: 10.1038/s41588-022-01018-x]. RCTD assumes that the observed spot level gene counts follow a Poisson-log-normal mixture. The mean of the log-normal distribution for the library size-normalized Poisson rate parameter is modeled with cell type-specific mean expression profiles, while accounting for platform effects by including a gene specific random effect term. It first uses external scRNA-seq reference data to estimate the mean gene expression profile of each cell type. The inferred platform effects are plugged into the probabilistic model to obtain the maximum likelihood estimates (MLE) of cell-type proportions. Thus, the deconvolution method employed a list of important gene expression of given cell types and given spots, not only the top marker genes, which leads to some level of inconsistency.

Indeed, the marker genes expression were not restricted to the location of posterior floor plate, but showed some expression in other sparsely distributed spots. The posterior floor plate is a unique structure and is well recognized in the small region that near the neural tube. Referring to the anatomical atlas (Kaufman's atlas of mouse development, 2015), the posterior floor plate cell type localization is correct. We reasoned that the inconsistency between cell type location and marker gene expression is mainly due to the different detecting ability of single-cell and ST. ST gives us a relatively complete view of gene expression across the whole embryonic tissues but single-cell sequencing may be confounded by cell selection. We also noticed that *Shh*, *Ntn1* showed reported expression in other tissue regions (e.g.:doi: 10.1093/hmg/8.12.2335-a for *Shh* expression; doi: 10.1016/j.ydbio.2011.04.016 for *Ntn1* expression in other tissues), which corroborate our ST data. We modified our description in Line 450 - Line 452 of the result part in the revised manuscript.

2. In Extended Data Fig. 3d, the authors show D14, D3 and D4 domains top regulons AUC activity score, while the top regulons TF expression on the right seems didn't match well. This should be explained or using better examples.

Our response: we thank the reviewer for the comment. We agree there are some extents of inconsistency between the activity of TF regulon and the expression of TF itself. Since the activity of a given regulon in each spot was calculated based on the expression levels of all the TF-target genes in this regulon, the activity score represents the enrichment level of all the TF-target genes, thus its level could be slightly different compared to the TF expression itself.

3. In the second paragraph of the section ‘Construction of a spatial transcriptome atlas of embryo organogenesis at E13.5’, the authors wrote, ‘For example, D6-hepatic parenchyma, specifically expressing Afb and Apoa2’. Afb should be Afp.

Our response: We sincerely thank the reviewer for the careful reading. This typo has been corrected.

4. In Figure 3a, the cell color inside Cluster 8 is inconsistent resulting in the different cell color in the left two most sections in figure 3b.

Our response: We apologized for the inconsistent color code in this figure and now we have amended it.

Reviewer #2 (Remarks to the Author):

In this study, Qu and colleagues performed spatial transcriptomic assays on 10 sections of mouse embryos staged at E13.5, to generate a 3D atlas. Leveraging this dataset, they next identified 19 distinct spatial domains globally and their key TFs, and then focused on several interesting questions regarding development, including subdomains of the visceral organs, and A-P/D-V axis of spinal cord. Moreover, they integrated the dataset with other single-cell RNA-seq datasets to further deconvolute the cell-type heterogeneity of each spatial spot. Although we find that study interesting, we do have a number of major concerns:

Our response: We thank the reviewer’s critical comments.

Major comments

1) There are several concerns regarding the “resolution” of the current spatial transcriptomic dataset:

- Most of the spatial domains (D1-D19) were defined in an “organ” or “tissue” scale, e.g. heart. Then, authors nominated key TFs or genes for individual spatial domains. However, the resolution might not be sufficient. People could perform RNA-seq on dissected tissue samples, or single-cell RNA-seq, to obtain the similar transcriptional information. We don’t see a significant improvement by performing spatial transcriptomics documented.

Our response: This comment is important. Though not in sing-cell resolution, our spatial atlas of mouse organogenesis provided much richer information than (single-cell) RNA-seq on dissected tissue samples from the aspects of spatial distribution of gene expression/TF regulon activity, comparable expression among multiple organs and spatial-aware of cell-cell communications. Furthermore, the transection view of spatial gene profiling provides us a unique angel to look into the D-V, L-R, A-P axes in the setting up the embryo. Take the spinal

cord development as an example, we revealed the spatial patterning of craniocaudal, dorsoventral and radial axes in spinal cord development, which would be hard to tackle by employing only RNA-seq on dissected tissue samples or with single-cell RNA-seq.

In addition, our dataset has extendable value for further sub-organ level analysis. As suggested by the reviewer, in the revised manuscript we focused on the developmental heart to achieve a higher resolution view. By subclustering the heart-associated spatial spots, we were able to dissect the molecular anatomical structure of heart on our spatial dataset into Atrium, Ventricle, Epicardium and Outflow Track (OFT). We identified the regional specific gene expression and also showed the consistent distribution of heart specific cell types in this region by integration analysis of ST and single-cell data. Besides, we identified a set of spatial co-expressed Congenital Heart Disease (CHD) related genes, and established their spatial location with the disease in the sub-heart region.

Moreover, we have added 4 more sections of spatial transcriptome from anterior to posterior (F6.5, F7.5, F8.5, and F9.5), and the integration analysis of all 14 sections showed that all spatial domains were identified from the represented 10 sections dataset. Taken together, this spatial transcriptomic dataset provides rich information about the critical gene expression and regulation, as well as spatial cell distribution and communication which are indispensable for the formation of complex tissue architecture.

Although the spatial atlas in this study is not at single-cell resolution, it provides a framework to map the single cells to a spatial coordinate (as also showcased in our study), which is a valuable resource for single cell studies.

- Most of the spatial domains or subdomains were purely identified by transcriptional heterogeneity. Do the author have anatomical features to verify those annotations? e.g. some type of imaging data.

Our response: Thanks for raising this question. We have collected the images of the given slides by 1% Cresyl violet staining before processing for spatial transcriptomics using Visium Spatial Gene Expression Kit. All these reference image data were organized and deposited on our website (www.most.ccla.ac.cn) and they were also publicly accessible through the deposited data under NGDC accession number (OEP003721).

We identified the spatial domains and subdomains based on the transcriptional heterogeneity unbiasedly, which are widely used to define the cell identities. Indeed, these domains and subdomains were carefully annotated by applying several complementary approaches: 1) we examined the expression of signature genes and enriched GO terms. For most biological systems, there is a scientific consensus on the genes expressed by particular cell types and the annotation based on this works well in a lot of practices; 2) we also double checked the spot identities based on the deconvolution analysis from single-cell data; 3) we did verify the molecular spatial structures with the spatial anatomical structure of the collected image data by referencing annotation of the Emouse Atlas, as shown in the right panel of Fig. 3b and the bottom panel of Fig. 3e. The identified subdomains by spatial transcriptome matched well with the annotation of the anatomical structure curated by experienced experts.

- The authors indicated “3D atlas” many times in the manuscript, however, I didn’t see any

analysis that was across different sections. It's not clear to me why the authors termed it 3D as opposed to just a limited set of sections. 3D implies a model that is really contiguous in all three dimensions.

Our response: We thank the reviewer for raising this concern. We collected 10 sections from a mouse embryo evenly and applied spatial analysis on all sections. To demonstrate the utility of this whole-10-section dataset, we analyzed the spinal cord region in Anterior-Posterior axis, which were across the head to tail and covers sections from 2 to 10 in our dataset. Within spinal cord regions, we also analyzed the spatial heterogeneity across dorsal to ventral, and along radial axes from the inner to outer (revised Fig. 4, Extended Data Fig. 8 and 9).

To intuitively visualize the spatial characteristics, we reconstructed a 3D embryo model through inserting these 10 sections into the spatial template model to illustrate the spatial domain and spatial expression/ regulon activity in a 3D space. We apologize for the error of our website server, which led to the unreachable of the mentioned 3D illustrations during the time of reviewing. Now we fixed the website server and also included the 3D embryo models into the Supplementary Video files.

As the reviewer mentioned that our dataset lacks real contiguous sections. Currently, it is really a big challenge to approach the real 3D model because of large size of embryo at this stage and limitation of available ST technology. We thus sampled 10 representative sections from the whole embryo. After careful data analysis and annotation, we observed that these collected sections covered most of the organ and tissue structures. To further demonstrate the coverage of collected sections, we performed 4 more sections from a female embryo during this revision, which were sampled about 50 sections away from respective sections of the male embryo. According to the analysis of newly added sections, we identified the similar gene expression patterns and spatial domain distribution between these two embryo datasets (revised Extended Data Fig. 3). Thus these 10 sections could be served as a core dataset and used to align other data to the embryo.

2) The authors identified the key TFs and TFs modules for individual spatial domains. Some potential concerns:

- It would be better to benchmark those TFs or TF regulons in some ways. At least authors should show which TF is known, which TF are relatively new, and which are unexpected.

Our response: We appreciate this suggestion. Through literature review, we classified the TFs into known or relatively new according to the related terms of spatial domain. For the reported TFs, we included represented references by listing the PMID in the Supplementary Table3. We also modified the descriptions in the results part accordingly and marked the relatively new TFs with bold in Extended Data Fig.5d which displayed the heatmap expression of corresponding TFs in spatial domains .

- The TF modules are very interesting, but it necessarily needs to dig deeper. Are those TF-TF interactions reliable or not? What are the potential molecular mechanisms (one or two examples)? Otherwise I don't think it's very informative compared to the left sections of the manuscript. Currently, without any experimental validation or literature supporting, all the

conclusions are more likely speculated from kinds of “co-expression” analysis.

Our response: We thank the reviewer for this important question. We agree the main purpose for the identification of TF-TF module here is to describe the phenomenon that there are some highly associated regulons which could form a module, and they may function in concert in establishing particular spatial domains. As is known that TF often work in combination to coordinate gene expression levels. Here, the TF modules were identified based on the TF regulon activity. TF regulons in each module suggested that these TF regulons may show potential co-activation in specific organs or tissues.

To further demonstrate the TF-TF association within each module, we applied two different approaches: 1) we performed protein-protein interaction network analysis. As the result showed below (Figure R3), significant connections of TFs in M1, M2, M6 and M7 were identified; 2) Due to the small size of M3(14 TFs), M4(5 TFs), and M5(5 TFs), we analyzed these TFs by CoCiter [doi: 10.1371/journal.pone.0074074.], which is a tool to infer gene function by assessing the significance of literature co-citation, we used the gene and term function and found significant connection of M3 with term “skin” or “limb” (p-value 0.004), M4 with “ganglion” (p-value 0.002), and M5 with “blood vessels” (p-value 0.02) or “Mesenchyme”(p-value 0.004), thus support the association of regulon module with top selected spatial domains.

Figure R3. The Protein-Protein interaction network analysis through STRING database for Modular M1, M2, M6 and M7.

- I feel the second half of the manuscript, including the subdomains of the visceral organ, A-P/D-V axis of spinal cord, and integrating with other datasets are way more interesting. I would suggest slightly reorganizing the manuscript to emphasize the second half.

Our response: Thanks for the suggestion. The figures and manuscripts were substantially revised. We have added more experimental validation and data analyses in the revised manuscript. Specifically, we reorganized and enhanced the subdomain analyses by performing subdomain characterization for heart (Fig. 3 and Extended Data Fig. 7), and performed more data validation for A-P/D-V axis of spinal cord (Fig. 4 and Extended Data Fig. 8). We also explicitly included sections from a female embryo and extracted the gonad-specific spatial features (Fig. 3, Extended Data Fig. 3 and 6).

3) A suggestion, in Fig.1d, is it possible to make a 3D visualization of all the 10 sections even though it's not in a "continuous" way? For example, domain 14 is distributed in three different sections, it might be hard for the audience to build a "3D" view of how domain 14 is distributed in a real whole embryo. I imagine something like the one in the subpanel (1) of the panel (a) would be much more straightforward.

Our response: We thank the reviewer for this suggestion and we're sorry that our website was not accessible during the review period and therefore we were not able to show the 3D embryo model. In the revised manuscript, we included the 3D embryo models with the distribution of each spatial domain in the Supplementary Video files just as the reviewer suggested. In the website, we not only illustrated a 3D visualization of spatial domains in a real embryo 3D model, but also showed spatial expression of all measurable genes and regulon activities across the 10 sections in the 3D space. As an example, we showed a 3D model illustrating the spatial distribution of Domain 14 in a virtual whole embryo and the spatial distribution of top marker gene *Foxg1* and top regulon *Neurod2* in 3D space (Figure R4).

Figure R4. A. Spatial distribution of Domain 14 in 3D embryo model. The spatial distribution of top marker gene *Foxg1*(B) and *Neurod2* (C) in 3D space.

4) Most of the identifications are way "safe". It doesn't sufficiently show how useful such an interesting dataset is.

Our response: Thanks a lot for this critical comment. Indeed, we identified many genes,

regulatory patterns, spatial domains which are already known by the previous studies. There are two obvious reasons: 1) the mouse development at this stage has been a long-lasting topic and many marker genes and organ-specific developmental mechanisms have been reported. Not surprising that many of them were echoed in our analyses. 2) As a resource-centered work, we anticipate that many novel findings will be thoroughly investigated in the future or by experts that are interested in the specific organs or pathways. However, we also identified and validated some relatively new genes, such as those in the spinal cord region. Nevertheless, according to the reviewer's comments, we revised our manuscript thoroughly to demonstrate the utility of the dataset. We summarized a few major conclusions as the following:

- a) We uncovered critical and relatively new genes and regulators in organogenesis for cell fate specification by spatial domain and sub-domain analysis and validated our findings by replicates of spatial transcriptomic data from 2 another embryos (Figure 3, Extended Data Fig. 1,3, 6 and 7), by experimental validations such as smFISH and whole mount ISH or through literature reviewing (Extended Data Fig. 2 and 8) .
- b) We revealed the spatial patterning and regulation of craniocaudal, dorsoventral and radial axes in spinal cord development and validated the expression of these spatial pattern genes by smFISH (Figure 4 and Extended Data Fig. 8).
- c) Through cell type spatial mapping, we illustrated the important cell colocalized intercellular communications and validated the spatial proximity of LRs by smFISH RNAscope (Figure 6 and Extended Data Fig. 13).
- d) To fulfill the goal of further deciphering the development of mouse organogenesis, we set up an expandable web portal for interested researchers to visualize the spatial domains in a whole embryo model and explore spatial gene expression and spatial activity of regulons in 3D space, we also provided a pattern search for finding the genes with similar spatial patterns.
- e) Our spatial analysis in heart region dissected the detailed spatial structures and signatures of heart and illustrated the spatial distribution of heart specific cell types. We also identified a set of spatial co-expressed coronary heart disease (CHD) related genes, and their spatial locations in the sub-heart regions.
- f) Taking spatial domain of heart as an example, we demonstrated the extensibility of our dataset in the detailed analysis of available organ or tissues including lung, liver, GI system with Gut, pancreas, stomach, gonad, brain and nerve systems with spinal cord, ganglion, cartilage and chondrocyte, ears and craniofacial region, and so on, which were not reported in a whole embryo system previously. And we also showed our data is compatible to other datasets collected by different methods such as TOME single-cell dataset (Figure 5 and Extended Data Fig. 10, 11, 12) from Cao et al. [doi:10.1038/s41588-022-01018-x] and MOSTA by spatial transcriptome (Extended Data Fig. 4) [doi:10.1016/j.cell.2022.04.003].

5) Could the author try to compare the current data to some other spatial transcriptome data during mouse organogenesis, e.g. MOSTA or sci-Space?

Our response: We thank the review for this important question. We have now combined our spatial data with representative section of MOSTA dataset at stage E13.5. We integrated the MOSTA with our data and applied label transfer to the MOSTA data. Interestingly, spatial

domains of our dataset matched relatively well with MOSTA annotation as shown in Extended Data Fig. 4, which indicates that our spatial dataset is compatible to other spatial datasets. We also showed that our spatial dataset has unique advantages because of the transverse section and more much sections that were collected. For example, we obtained organ or tissues which were not included in MOSTA, such as gonad, ganglion, and stomach at this developmental stage.

Minor comments

1) In Fig. 2a, only some of the TFs are highlighted on the right of the heatmap. How were they selected?

Our response: Sorry we didn't describe it clear. We manually checked the function of transcription factor for each regulon and then selected those which were well known to be associated with the corresponding spatial domain or relatively new. We didn't list all of them because of the space limitation. The whole list of regulons associated with each spatial domain were provided in Supplementary Table 2. We also manually classified the top regulons shown in the heatmap into known or relatively new in Supplementary Table 3 .

2) In Fig. 2a-b, I am curious, does higher RAS of TF regulon also guarantee higher TF expression? It would be interesting to plot gene expression for those TFs as well, i.e., making the same heatmap but using TF expression instead of RAS.

Our response: Thanks for this suggestion. We have added the TF expression heatmap in the Extended Data Fig. 5d and it showed that the average activity of regulons and the average expression of TF in each spatial domain matched well. In our previous submission, we plotted the spatial distribution of TF of the respective regulon with top 1 RAS in spatial domain as showed in Extended Data Fig. 5e. Generally, these TF expression have good match with the activity score of TF regulons. However, according to the AUCscore algorithm which is implemented for calculating the RAS, it takes into account the expression of all the TF-target genes which comprise a TF regulon. Hence, the spatial activity of regulons showed a more represented expression compared to the TF expression. Therefore, the TF regulon tends to highlight the coherent TF-target expression which convey some kind of biological regulation.

3) It's not clear to me - "By computing the averaged RAS score for each TF regulon module ... in particular tissue regions" (Page6, 182-184). How were the region-specific modules of TF regulons identified? More specifically, in the Fig.2c, how was each module assigned with one or more spatial domains?

Our response: Sorry for the confusion. The region-specific modules of TF regulons were identified mainly based on the averaged regulon activity scores of all regulons in given modules (Fig.2c the right most panel). We didn't apply a threshold here, if spatial domains show very high activity score for a given regulon module, these spatial domains are selected, as shown in

the violin plot below (Figure R5). For example, in M2, the spatial region of D6-hepatic parenchyma showed very high activity scores, so this module was classified as liver related regulon module. As a proof of the robustness, we looked into the association between the TFs contained in the module and the corresponding spatial domains. Regulons such as Hnf1a and Nr1h4 belonging to this module were top regulons that associated with liver (D6) revealed by Regulon Specific Score (RSS) (Fig. 2b). It doesn't mean that only the selected domains are associated with this module but just mean these domains show potential higher association based on the spatial distribution of their activity scores.

Figure R5. Violin plot of the average activity score for modulars in each spatial domain.

4) In Fig. 4a-b, the colormap is not quite distinct for different sections, S2-S10. I suggest using a different colormap. Also, Hox genes are very specific to the anterior and posterior, but they don't show a clear pattern in the central region, could the authors comment on it?

Our response: Thanks a lot for this suggestion. We now updated the colormap for different sections (Fig. 1B and Fig. 4a-b and Extended Data Fig. 8 and 9).

As for the Hox expression, the Hox family genes showed a concrete patterning along the A-P axis. For example, Hox2-5 are specific to the anterior, and the Hox 9 -10 are mainly specific to the posterior regions, and Hox 6-7 and Hoxb9 are more specific to the central regions. As more evidenced in the gene expression heatmap, Hoxb6 showed a higher expression along section 2-4, and Hoxac8 with a higher expression along section5 to 6, corroborating the established Hox code in setting up the body axes [doi:10.1101/gad.303123.117].

Reviewer #3 (Remarks to the Author):

Mammalian embryogenesis is a well-orchestrated process involving a series of cell proliferation and differentiation processes. Understanding the cellular dynamics and spatial distribution of developing cells during organogenesis is critical in developmental biology. In this study, Qu et al used 10X Visium spatial transcriptomics technology to analyze cross sections of an E13.5 stage mouse embryo. The authors carried out extensive exploration of the spatial patterning of gene expression during embryogenesis, including tissue type characterization, TF network and regulation inference, A-P axis expression dynamics, and cell type deconvolution. Meanwhile, the authors proposed to use Visium spot as the natural aggregation of different cell types to test for cell-cell communication through ligand-receptor pair analyses. Overall, the computational analysis is comprehensive and well-executed, though the study itself still has some major weaknesses, detailed below.

Our response: Thanks for the acknowledgement of our spatial transcriptomic study and the computational analysis.

First, the whole work was based simply on 10X Visium experiments using 10 tissue sections, out of ~1000 sections from a single mouse embryo at E13.5. The highly selected tissue sections indicate strong sampling bias in projecting spatial gene expression patterns for the embryo. It is almost impossible to reconstruct a 3D transcriptomic model of the E13.5 embryo. For this reason, it is very hard to call this dataset a '3D' spatial transcriptomic atlas/resource, nor being truly useful for the community. The authors did validate the data quality with a second embryo with two sections. However, this validation data does not resolve the intrinsic bias in the main data, let alone the poor quality in the second embryo. To resolve this sampling bias, the easiest solution is to unbiasedly increase the number of embryo sections to the level that could enable the reconstruction of a 3D transcriptomic model.

Our response: Thanks for the critical comments. We collected 10 sections from a mouse embryo evenly and applied spatial analysis on all sections. To demonstrate the utility of this whole-10-section dataset, we analyzed the spinal cord region in Anterior-Posterior axis, which were across the head to tail and covers sections from 2 to 10 in our dataset. Within spinal cord regions, we also analyzed the spatial heterogeneity across dorsal to ventral, and along radial axes from the inner to outer (revised Fig. 4, Extended Data Fig. 8 and 9).

As the reviewer mentioned that it is a big challenge to reconstruct a 3D transcriptomic model of the E13.5 embryo because of large size of embryo at this stage and the limitation of available ST technology. Although we have demonstrated that our collected sections covered most of the organ and tissue structures after careful data analysis and annotation, we performed new experiments by collecting spatial transcriptome of 4 new sections from a female embryo in the revised manuscript to further prove the representation of these sections. The collected positions were sampled about 50 sections away from respective sections of the male embryo which we named as F6.5, F7.5, F8.5 and F9.5. By integrating these two-embryo dataset together without applying any batch removal methods, we found these two datasets were highly compatible, and matched well with each other as shown in Extended Data Fig. 3. Thus, we applied label transfer

to annotate the spatial regions, the annotation of the spatial domains matched well with their spatial anatomic structure across different embryos (Extended Data Fig. 3). These results showed that our dataset with 10 sections were a good representation of the major organ or tissues of an embryo without significant sampling bias, hence could be served as a core spatial atlas at the mid-stage of mouse organogenesis.

To intuitively visualize the spatial characteristics, we also built a 3D embryo model through inserting these 10 sections into the spatial template model to illustrate the spatial distribution of spatial domains. The gene expression of whole genome and regulon activity were also displayed in a 3D space. Thus, albeit lacking sufficient coverage of a whole embryo, our data is representing the distal-proximal dimension of the embryo to some degree. Please also kindly refer to our responses to similar concerns by the reviewer 2.

Second, despite very comprehensive analyses, the bulk of the results are descriptive and just validate what is already known in the specific areas. Regardless, the data could have led to non-descriptive new discoveries. The study does not generate new biological discoveries that could potentially demonstrate the power of ultimately building a comprehensive and single-cell resolution 3D spatial transcriptomic atlas. The authors claimed to discover spatial patterns of new genes – which are supposed to be expected – these results should be backed up with more solid validation with e.g. whole embryo ISH assays and/or lacZ reporter assays, and additionally examine their biological function in case studies. These validations are more necessary considering that Visium ST technology only analyzes mini-bulk transcriptomes, and bears strong technical noise including leaking transcripts across the section.

Our response: We thank for these critical comments. In this work, we performed a holistic spatial transcriptome atlas of all major organs at mid-organogenesis stage (E13.5) of mouse embryo and delineated a 3D rendering of the molecular regulation of embryonic patterning by system sampling 10 sections from anterior to posterior. This is a comprehensive molecular architecture of the embryo development at mouse mid-organogenesis with deep gene detection (a median of 5,668 genes) and covering most of organs and tissue. Compared to other spatial data, for example the MOSTA, our spatial dataset included some organs or tissues which were not included in MOSTA at this stage, such as gonad, ganglion, stomach and so on.

Secondly, the regulation network analysis showed some novel regionalized specific regulons, which will be beneficial to the embryonic biologists who are interested in exploring the regulatory mechanism across different cell types and locations in controlling embryo organogenesis.

Thirdly, the craniocaudal, dorsoventral and radial axes patterning of spinal cord were revealed and spatial genes were identified to constitute the regionalized patterning in embryogenesis.

Finally, we developed a spatial cell-cell communication (CCC) analysis workflow (named STcomm) to systematically infer the cell-cell communication under the spatial context, and we verified that *Nrxn3-Nlgn1* showed prevailing communication probability between the neuron progenitors and the inhibitory interneurons.

We now added more experimental validations to confirm the spatial patterns of the new genes by Whole mount In Situ Hybridization (WISH) assay and smFISH RNA-scope assay as shown

below and in Fig. 4f, 4h and Extended Data Fig. 2e,2f, 8c. Furthermore, we also thoroughly checked the literatures for the identified marker genes and confirmed that a large proportion of genes identified as spatial marker genes were validated in RNA level by WISH or protein level by immunostaining. They were reported to play important functions during development through biology functional studies. For example in the skin domain, we found that among the top 20 genes of skin domain, 18 genes were reported as key genes for the skin development during organogenesis and only two were relatively new. We thus performed whole amount analysis on one gene as shown in the Figure R6 and Extend data Fig. 2, which confirmed its specific expression in skin.

However, constrained by the scope of this study, the reporter assay or functional examination by gene trapping for example, is waiting for further in-depth investigation. We hope our data could be a useful resource for developmental biologists who are interested in specific organ development. To fulfill this purpose, we have provide all the gene expression and regulon activity in the web portal (www.most.ccla.ac.cn). The data can be accessed in an interactive and easily accessible way.

Figure R6 Whole mount In Situ Hybridization of domain specific genes in E13.5 embryos.

A, *Sfn5* is a relatively new marker in skin development which is identified from D15_Skin domain, and the WISH showed that *sfn* high expressed in whisker hair follicles. B, *Pantr1* is the top gene in D14-Neopallium with less study in brain development, and the WISH showed that *Pantr1* high expressed in the cortex.

Third, what is the sex of this embryo? This is critical information for mouse embryos developed beyond E11 when the gonad starts to have sex-specific differentiation. The authors claimed in Line 262-264 that *Lefty2*, *Tex19.1*, and *Dppa3* are gonad-specific genes (Fig. 3c). Are they male- or female-specific gonad genes? It will be interesting to see the expression pattern of sex-specific gonad gene expression, such as *Sox9*. For this reason, the authors should analyze both male and female embryos for a comprehensive resource, and again, could also use developing gonads as a system to demonstrate the power of ST technologies for studying developmental biology questions.

Our response: We thank the review for raising these critical questions. We have incorporated

these suggestions in the revised manuscript. The 10-section is from a male embryo (Revised Extended Data Fig. 3c, d). During the revision, we performed spatial transcriptome for 4 more sections from a female embryo (Revised Extended Data Fig. 3). And the section 9.5, a similarly positioned section from the male embryo, contains the gonad by examining the spatial anatomic structure. As mentioned above, the integration analysis of the new 4 sections showed well compatible with the reference dataset. We thus focused our analysis on the sex difference of gonad and identified sex-specific gonad maker genes between male and female such as steroid hormone genes *Cyp17a1* [doi: 10.1016/j.mce.2021.111261], *Wnt4* in sex determination [doi: 10.1016/j.biocel.2006.06.007], and also gonad common markers across sex, such as *Dppa3* and *Dazl* which are important regulator in pluripotency [doi: 10.1159/000074346 and doi: 10.1073/pnas.1910733116]. Interestingly, as the reviewer suggested, we identified that *Lefty2* are male specific gonad marker genes, and *Tex19.1* and *Dppa3* (and other *Dppa* family genes such as *Dppa4*) are common gonad marker genes which showed no sex difference. Meanwhile, *Sox9* reported as a key player in sex determination showed expression across organs but relatively low in female gonad. (Figure R7).

We would like to highlight that compared to methods of single-cell RNA-seq or bulk RNA-seq, the ST technology which relies on the visual inspection of tissue section, allow us to pinpoint the organ of interest precisely and quickly. Therefore, we are able to efficiently and confidently dissect the sex difference in a defined transverse section. All the above showcased the power of ST technologies.

Figure R7. A. The expression of Sox9 across male and female visceral organs. B. The spatial distribution of Sox9 in male and female gonad as indicated.

Other concerns:

The mouse embryo sections have been well-characterized. However the authors did not seem to use their H&E staining images for cross-reference analyses. Since the embryo is not straightly A-P aligned, the H&E staining helps bridge the actual tissue and the ST data. To the least, presenting the corresponding H&E staining images for each analyzed section helps to indicate tissue quality in the ST experiment. Fig. 1a schematic does not seem to reflect the actual sections used in this study.

Our response: We are sorry that we didn't describe it clear. The sections shown in Fig 1a were a schematic representation of the real sections. We optimized the workflow of 10X Visium to shorten the time and increase the detected genes according to our experience in spatial

transcriptomic assays of mouse embryo development. We have collected the image of the given slides by staining with 1% Cresyl violet solution instead of H.E. We have organized these image data on the website serve as a reference and they were also publicly accessible through the deposited data under NGDC accession number (OEP003721). We did verify the molecular spatial structures with the spatial anatomical structure of the collected image data by referencing annotation of the Emouse Atlas, as shown in the right panel of Fig. 3b and the bottom panel of Fig. 3e. The identified subdomains by spatial transcriptome matched well with the annotation of the anatomical structure curated by experienced experts.

Line 131, in the text, D17-ependyma was presented as ubiquitous existence across the assayed embryo sections. However, in Extended Fig. 2b, it was only evident in the head sections. The authors need to examine carefully if any misinterpretations of the results.

Our response: We are sorry for the oversight. Indeed, there were only 3 or 6 spots in section 4, section 6, section 7. We revised our description in the manuscript.

Fig. 2c, how was the value of molecular mean activity calculated? Meanwhile, D4, D5, and some other groups seemed missing in this analysis.

Our response: Sorry that we didn't describe it clear. The mean regulon activity scores of a given module is calculated by averaging the activity scores (RAS) of all regulons that belongs to this module in each spot. And we then visualized the mean activity of each module by spatial plot. The module associated spatial domains were identified mainly based on the mean regulon activity scores of modules (Fig.2C the right most panel). If spatial domains show very high activity score for a given regulon module, these spatial domains are selected. Some spatial domains such as D4 and D5 were not associated with any module, thus were not shown in the Fig. 2c.

A similar problem happened in Fig. 4a-c where sections 1&9 or 1&10 were missing.

Our response: Thanks for this question. Only section 2 to 8 and section 10 were annotated with D16 Medulla oblongata and spinal cord (Fig. 1c and Extended Data Fig. 2b). In the brain region of Section 1, there was no D16 Medulla oblongata and spinal cord spots. And the spinal cord region was also missing in Section 9 because of the incompleteness of Section 9. We have corrected the S9 to S10 in Fig. 4c to make the labels consistent among Fig. 4a-c.

Extended Fig. 1i and n, there seem to be typos in the tick labeling to distinguish E1 and E2.

Our response: We feel sorry for our carelessness. In the revised manuscript, the typos are corrected.

At last thank you for your constructive comments, which helped us to improve the rigorousness of our study and strengthen our findings.

REVIEWERS' COMMENTS

Reviewer #1 (Remarks to the Author):

The manuscript has been significantly improved and would make an informative resource for understanding mouse organogenesis.

Reviewer #2 (Remarks to the Author):

The revised manuscript has been significantly improved, and we are satisfied their response to reviewers' comments.

A small issue is that their website <http://www.most.ccla.ac.cn/> seems not to working.

Reviewer #3 (Remarks to the Author):

The reviewer thanks the authors for making the revision, resolving most of the questions. However, the reviewer is still not convinced that such a collection of 10 out of ~1000 slices of an embryo is representative to reconstruct a 3D transcriptomic map of a developing embryo. Although the authors detected all major organs, this is still a very preliminary representation of the actual developing embryo, at E13.5. To the least, the authors need to honestly discuss the limitations of the study, e.g. the missing spatial transcriptomic information from the current study, and in the meantime, tone down some of the claims in the manuscript.

Response to reviewers

Reviewer #1 (Remarks to the Author):

The manuscript has been significantly improved and would make an informative resource for understanding mouse organogenesis.

Our response: We thank the reviewer for the positive comments.

Reviewer #2 (Remarks to the Author):

The revised manuscript has been significantly improved, and we are satisfied their response to reviewers' comments.

Our response: We thank the reviewer for the positive comments.

A small issue is that their website <http://www.most.ccla.ac.cn/> seems not to working.

Our response: The website is available as <http://most.ccla.ac.cn>. We apologized for misspelling the address. We used the "naked domain" format, which means the domain name without the "www" prefix.

Reviewer #3 (Remarks to the Author):

The reviewer thanks the authors for making the revision, resolving most of the questions. However, the reviewer is still not convinced that such a collection of 10 out of ~1000 slices of an embryo is representative to reconstruct a 3D transcriptomic map of a developing embryo. Although the authors detected all major organs, this is still a very preliminary representation of the actual developing embryo, at E13.5. To the least, the authors need to honestly discuss the limitations of the study, e.g. the missing spatial transcriptomic information from the current study, and in the meantime, tone down some of the claims in the manuscript.

Our response: We thank the reviewer for the comments. We agree that indeed this study does not provide a full 3D reconstruction. We toned down the claims by indicating “at the whole embryo scale”, “a representation”, instead of “the whole embryo” and “complete”. We modified the description throughout the manuscript. Moreover, we specifically discussed the limitation of this study in the last paragraph at the first place “*It should be mentioned that the present study does not encompass a complete 3D reconstruction of an E13.5 embryo, as it would necessitate the examination of a substantially greater number of sections. Moreover, the spatial resolution of this atlas is still not at single-cell level. Therefore, specific cell types or micro-structure of organs may be hidden at the current scale.*”. We also emphasis that more

sections and stages are needed in the future to generate a 4D atlas by the following discussion
“We envision a compiling of more developmental stages with spatial transcriptomics on a large number of continuous tissue sections or even within the intact tissues to generate a 4D atlas will greatly deepen our understanding of mammalian embryogenesis and expedite directed generation of various organs in vitro.”

Some particular modifications are listed below:

We modified *“Therefore, a 3D spatial transcriptome profiling of tissue and organ specific microarchitecture to align serial sections of the embryo into an entire organism containing all major tissue types is vital for our understanding of embryo organogenesis, which has not been reported.”* to

“Therefore, it is crucial to profile the spatial transcriptome of tissue and organ specific microarchitecture by aligning multiple tissue sections of mouse organogenesis embryo to a 3D template, which has not yet been reported.”

At the Methods section:

We modified *“Web service and 3D reconstruction”* to *“Web service and 3D illustration”*.

We modified *“To reconstruct the 3D embryo model, the images of TS22 corresponding to stage of E13.5 were retrieved from eMouseatlas (www.emouseatlas.org), we then registered our 10 section images manually with the matched images in TS22 using Adobe Photoshop, and stacked all these images together by ImageJ.”* to

“To illustrate the molecular characteristic and spatial domains of E13.5 embryo at a three-dimensional whole embryo scale, we first retrieved the images of TS22 corresponding to stage of E13.5 from eMouse atlas (www.emouseatlas.org), and then registered our 10 section images manually with the matched images in TS22 using Adobe Photoshop, and stacked all these images together by ImageJ.”